# Coordinated regulation of gene expression in *Plasmodium* female gametocytes by two transcription factors

Yuho Murata[1], Tsubasa Nishi[1], Izumi Kaneko[1], Shiroh Iwanaga[2], Masao Yuda[1]*

[1]Department of Medical Zoology, Mie University School of Medicine, Tsu City, Japan; [2]Department of Molecular Protozoology, Research Center for Infectious Disease Control, Osaka, Japan

*For correspondence:
m-yuda@doc.medic.mie-u.ac.jp

**Abstract** Gametocytes play key roles in the *Plasmodium* lifecycle. They are essential for sexual reproduction as precursors of the gametes. They also play an essential role in parasite transmission to mosquitoes. Elucidation of the gene regulation at this stage is essential for understanding these two processes at the molecular level and for developing new strategies to break the parasite lifecycle. We identified a novel *Plasmodium* transcription factor (TF), designated as a partner of AP2-FG or PFG. In this article, we report that this TF regulates the gene expression in female gametocytes in concert with another female-specific TF AP2-FG. Upon the disruption of *PFG*, majority of female-specific genes were significantly downregulated, and female gametocyte lost the ability to produce ookinetes. ChIP-seq analysis showed that it was located in the same position as AP2-FG, indicating that these two TFs form a complex. ChIP-seq analysis of PFG in *AP2-FG*-disrupted parasites and ChIP-seq analysis of AP2-FG in *PFG*-disrupted parasites demonstrated that PFG mediates the binding of AP2-FG to a ten-base motif and that AP2-FG binds another motif, GCTCA, in the absence of PFG. In promoter assays, this five-base motif was identified as another female-specific *cis*-acting element. Genes under the control of the two forms of AP2-FG, with or without PFG, partly overlapped; however, each form had target preferences. These results suggested that combinations of these two forms generate various expression patterns among the extensive genes expressed in female gametocytes.

## eLife assessment

This study offers **important** insights into the transcriptional regulatory networks driving female gametocyte maturation in rodent malaria parasites. The work is based on **solid** methodology and shows how two female-specific transcription factors, AP2-FG and PFG (aka Fd2), cooperate to upregulate the expression of genes required for development after fertilization occurs in the mosquito midgut. This study will be of interest to scientists working on sexual differentiation and gene regulation in *Plasmodium* and other apicomplexan parasites.

## Introduction

*Plasmodium* gametocytes are erythrocytic sexual stages essential for parasite transmission to mosquito vectors. Within the blood meal of a mosquito, female and male gametocytes egress host erythrocytes and transform into gametes for fertilization. Zygotes undergo meiotic nuclear division and develop into motile ookinetes to escape from the blood meal to the mosquito midgut epithelium. The elucidation of gene regulation during this lifecycle stage is important for developing antimalarial strategies that block transmission and prevent the spread of the disease.

Gametocytes develop from a subset of asexual erythrocytic parasites that are committed to sexual reproduction. The transition from the asexual erythrocytic stage to gametocytes occurs in late trophozoites; a subpopulation of trophozoites differentiates into gametocytes for sexual reproduction, and the rest develop into schizonts to continue asexual proliferation. Early gametocytes display no sex-specific features, but after a period of time, which is different among *Plasmodium* species, they start differentiating into each sex and finally become mature male and female gametocytes that are ready to transform into gametes. In *Plasmodium berghei*, early gametocytes manifest morphologies distinguishable from asexual stages at 18 hpi (hours post-infection of the new red blood cell) and manifest sex-specific features at 20 hpi. They then complete a sex-specific maturation process, during which a sex-specific gene expression repertoire is established, at approximately 24–26 hpi (*Mons et al., 1985*). Sex-specific proteomic and transcriptomic analyses in *P. berghei* have revealed a number of genes specifically expressed in each sex (*Khan et al., 2005*; *Witmer et al., 2020*; *Yeoh et al., 2017*). In females, genes necessary for fertilization and protein synthesis are expressed, and the number of transcripts is stored in an untranslated state in the cytoplasm to prepare for zygote development (*Mair et al., 2006*; *Guerreiro et al., 2014*). In males, genes for flagella-based motility and genome replication are expressed, as well as genes for fertilization.

Recently, it has become clear that transcription factors (TFs) belonging to the AP2 family play a central role in sexual development. AP2-G is essential (*Sinha et al., 2014*). Disruption of this gene causes the parasite to lose its ability to generate gametocytes. In *P. berghei*, its expression starts from the late trophozoite at 14–16 hpi and peaks at 18 hpi when sexual stage-specific features first appear (*Yuda et al., 2021*). *P. berghei* AP2-G induces hundreds of genes that contain several TFs important for the development of gametocytes, suggesting that induction of these TFs by AP2-G generates a driving force for gametocytogenesis (*Yuda et al., 2021*).

AP2-FG is a member of the *Plasmodium* AP2 family of TFs and is a target genes of AP2-G (*Yuda et al., 2021*; *Yuda et al., 2019*). AP2-FG is a female-specific TF and plays an essential role in the development of female gametocytes. In *P. berghei*, *AP2-FG* is expressed from early gametocytes to mature females, and by disruption of *AP2-FG*, the development of the female gametocytes is impaired, showing immature morphologies and resulting in complete loss of capability to mediate parasite transmission to mosquitoes. This TF binds to a ten-base female-specific *cis*-acting element and regulates variety of genes including those for fertilization, meiosis, and the development of ookinetes. This broad repertoire of target genes and their essential role in female development suggest that this TF is a master regulator of female development. However, *AP2-FG*-disrupted parasites can produce ookinetes with decreased numbers and abnormal morphologies, suggesting that additional mechanisms for activating female-specific genes are still functional upon the disruption of *AP2-FG*.

We explored novel TFs for gametocyte development in the target genes of AP2-G and identified a novel female gametocyte-specific TF. In this article, we report that this TF, designated as PFG, regulates the gene expression in female gametocytes in concert with AP2-FG.

## Results

### *PFG* is a target gene of AP2-G and is expressed in female gametocytes

We screened for novel sequence-specific TFs among target genes that were functionally unannotated in PlasmoDB (https://plasmodb.org/plasmo/app/) using highly conserved amino acid sequences among *Plasmodium* species as a criterion. We hypothesized that the amino acid sequences of DNA-binding domains of sequence-specific TFs would have been difficult to change during evolution because even a small change in the sequence specificity of TFs could cause catastrophic effects by global changes in gene expression. Through this screening, a gene encoding a 2709 amino acid protein with two regions highly conserved among *Plasmodium* was identified (PBANKA0902300, designated as *a partner of AP2-FG* [*PFG*]; *Figure 1A*). This gene is one of the *P. berghei* genes that were previously identified as genes involved in female gametocyte development (named *FD2*), based on mass screening combined with single-cell RNA-seq (*Russell et al., 2023*). The two conserved regions comprised 108- and 139 amino acids, respectively, and each region showed 94 and 91% conservation between *P. berghei* and *Plasmodium falciparum* (*Figure 1B*). The Blastp search (https://blast.ncbi.nlm.nih.gov/Blast.cgi) using these regions revealed that genes homologous to *PFG* exist broadly among apicomplexan parasites and also in alveolates closely related to Apicomplexa, such as *Vitrella brassicaformis* (*Figure 1C*;

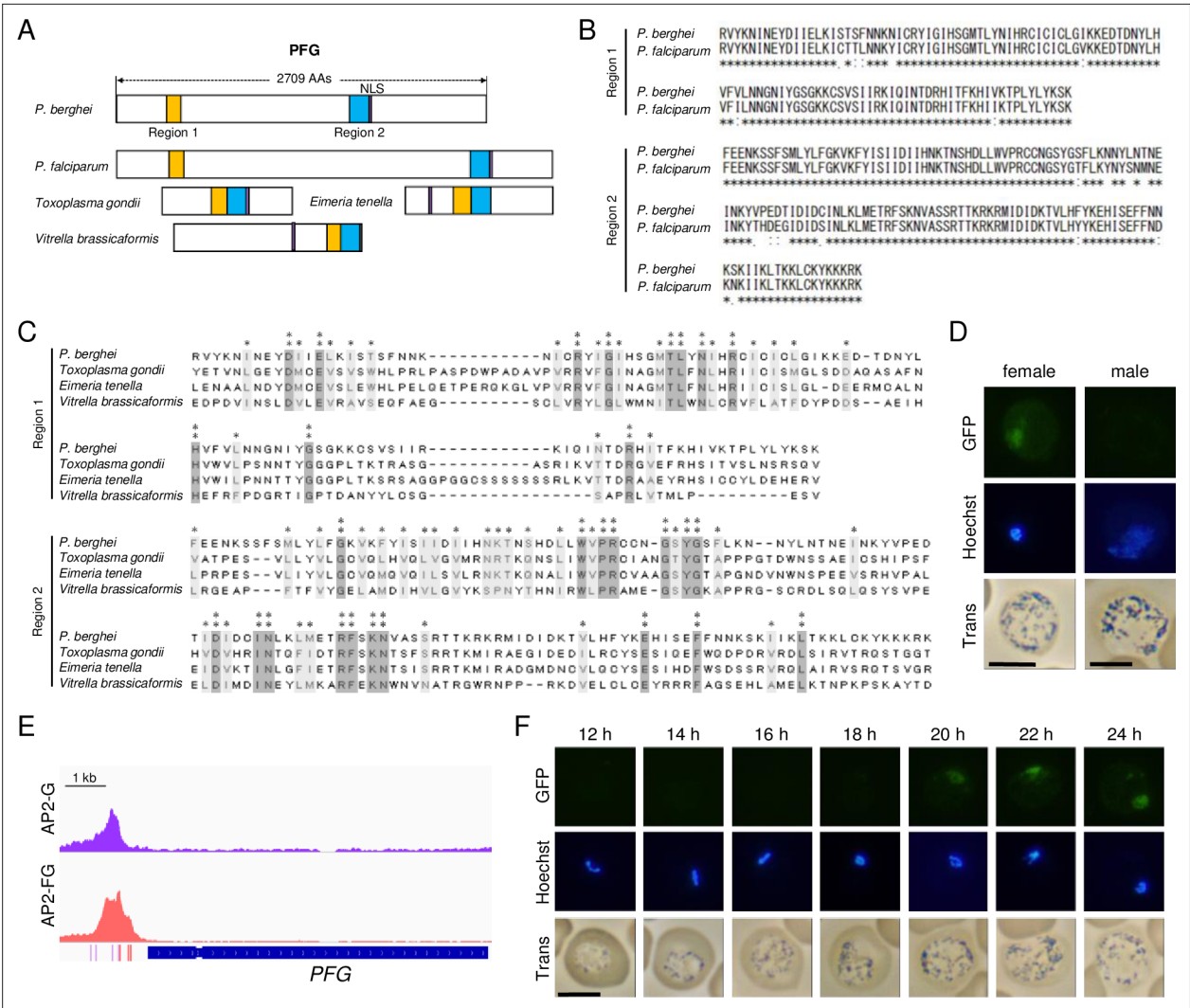

**Figure 1.** PFG is a target gene of AP2-G and is expressed in female gametocytes. (**A**) Schematic diagram of PFG from *P. berghei* and its homologs in apicomplexan parasites. Regions homologous to regions 1 and 2, which are highly conserved among *Plasmodium* species, are shown as yellow and blue rectangles, respectively. Nuclear localization signals were predicted using the cNLS mapper. The gene IDs of *P. berghei* PFG, *P. falciparum* PFG, and their homologs in *Toxoplasma gondii*, *Eimeria tenella,* and *Vitrella brassicaformis* are PBANKA_0902300, PF3D7_1146800, TGGT1_239670, ETH2_1252400, and Vbra_10234, respectively. (**B**) Alignment of amino acid sequences of regions 1 and 2 from *P. berghei* and *P. falciparum* PFGs was performed using the ClustalW program (https://www.genome.jp/tools-bin/clustalw). (**C**) The amino acid sequences of regions 1 and 2 from *P. berghei* PFG and its homologs from other apicomplexan parasites in (**A**) were aligned using the ClustalW program in MEGA X. The positions at which all these sequences have identical amino acids are indicated by two asterisks, and positions with amino acid residues possessing the same properties are indicated by one asterisk. (**D**) Expression of *PFG* in mature male and female gametocytes of *PFG::GFP* parasites. Nuclei were stained with Hoechst 33342. Scale bars: 5 µm. (**E**) Integrative Genomics Viewer (IGV) images from the ChIP-seq data of AP2-G and AP2-FG in the upstream region of *PFG*. Data were obtained from previous papers (*Yuda et al., 2021*; *Yuda et al., 2019*). The purple and red bars indicate the binding motifs of AP2-G and AP2-FG in the peak region. (**F**) Time-course observations of *PFG* expression during gametocyte development using *PFG::GFP* parasites. Parasites were observed every 2 hr from 12 hpi onwards. Nuclei were stained with Hoechst 33342. Scale bar: 5 µm.

The online version of this article includes the following source data and figure supplement(s) for figure 1:

**Figure supplement 1.** Preparation of PFG::GFP parasites.

**Figure supplement 1—source data 1.** Original gel image for *Figure 1—figure supplement 1*.

*Oborník et al., 2012*), whereas a large inter-region between these two regions present in all *Plasmodium* homologs was not observed in these proteins (*Figure 1A*). Based on domain prediction using the SMART database (http://smart.embl-heidelberg.de/), no functionally characterized domains were identified to be significantly similar to these two regions.

To investigate the expression of *PFG*, we generated parasites expressing GFP-fused PFG (*PFG::GFP* parasites). Fluorescence analysis showed that this gene is expressed only in female gametocytes and the protein is localized in the nucleus (*Figure 1D*), suggesting its involvement in the transcriptional regulation of female gametocytes. According to sex-specific transcriptome data (*Witmer et al., 2020*), the PFG is specifically transcribed in female gametocytes. We examined the target genes of the female-specific transcriptional activator AP2-FG and found that PFG had been predicted as a target of AP2-FG and harbored AP2-FG peaks with binding motifs in the upstream region (*Figure 1E*). These results suggest that *PFG* was activated in two steps, that is, by AP2-G and then by AP2-FG, during female development. Temporal profiling of the expression of *PFG* using *PFG::GFP* parasites showed that the expression became visible from 20 hpi (*Figure 1F*). The timing of the expression was approximately 4 hr later than that of AP2-FG, which started at 16 hpi (*Yuda et al., 2019*).

## PFG is essential for female development

We generated *PFG*-disrupted (*PFG*(-)) parasites to assess the role of *PFG*. The asexual stages of *PFG*(-) parasites proliferated as wild-types (*Figure 2A*), and morphologically mature male and female gametocytes were observed on Giemsa-stained blood smears (*Figure 2B*). However, in mosquito transmission experiments, no oocysts were observed on the mosquito midgut wall 14 d after the infective blood meal in two independent clones (*Figure 2C*). In ookinete cultures, only approximately 20% of the female gametocytes were converted into zygotes, and no banana-shaped or retort-form ookinetes were generated (*Figure 2D*). Considering the female-specific expression of *PFG*, these results suggest that disruption of *PFG* significantly impaired the development of female gametocytes. In the cross-fertilization experiment, ookinetes were formed when crossing with normal females (*P48/45*(-)) but not when crossing with normal males (*P47*(-)), demonstrating that the abnormal phenotype observed in *PFG*(-) was indeed derived from female gametocytes (*Figure 2E*).

Next, we generated *PFG*(-) parasites from transgenic *P. berghei* ANKA 820 cl1m1cl1 (here called 820) parasites, which express red fluorescent protein (RFP) under the control of the female-specific promoter (*CCP2*) and GFP under the control of the male-specific promoter (dynein gene) (*PFG*(-)$^{820}$ parasites) (*Raine et al., 2007*; *Mair et al., 2010*). Although female gametocytes with mature morphology were observed on Giemsa-stained blood smears as the original 820 parasites, in fluorescence-activated cell sorting (FACS) analysis, no RFP-positive parasites were detected in the *PFG*(-)$^{820}$ parasites (*Figure 2F*). In contrast, GFP-positive parasites were observed in them as in the original 820 parasites (*Figure 2F*). These results indicated that the promoter activity of the female-specific CCP2 gene used for RFP expression in 820 parasites was severely reduced by the disruption of *PFG*.

## Female-specific genes are globally downregulated in *PFG*(-) parasites

The above results suggest that *PFG* encodes a TF that plays an important role in female-specific gene expression. To investigate the effect of *PFG* disruption on transcriptional regulation in female gametocytes, we performed RNA-seq analyses of the wild-type and *PFG*(-) parasites (*Supplementary file 1*). Mice were treated with phenylhydrazine prior to the passage of parasites, and after infection, asexual stages were killed by treatment with sulfadiazine in drinking water. Three independent samples were prepared from wild-type and *PFG*(-) parasites. In *PFG*(-) parasites, 279 genes were significantly downregulated ($\log_2$(fold change) $< -2$ and p-value adjusted for multiple testing with the Benjamini–Hochberg procedure <0.001), more than 90% of which were female-specific genes according to sex-specific transcriptome data (*Witmer et al., 2020*; *Figure 2G*). Consistent with the results of the gene targeting experiment in 820 parasites, transcripts of the CCP2 gene decreased over 20 times in *PFG*(-)parasites. Genes significantly downregulated in *PFG*(-) constituted more than half of the female-enriched genes (258/504, *Supplementary file 1*), suggesting that the expression of female-specific genes was globally downregulated in *PFG*(-) parasites.

## PFG is co-localized with AP2-FG on ten-base motif

To examine whether PFG targeted these female-specific genes, ChIP-seq analysis was performed using *PFG::GFP* parasites and anti-GFP antibodies. In two biologically independent experiments, experiments 1 and 2, 1073 and 1204 peaks were identified in the genome, respectively (*Supplementary file 2a and b*), and 95% (1029 peaks) of the peaks identified in experiment 1 were common with

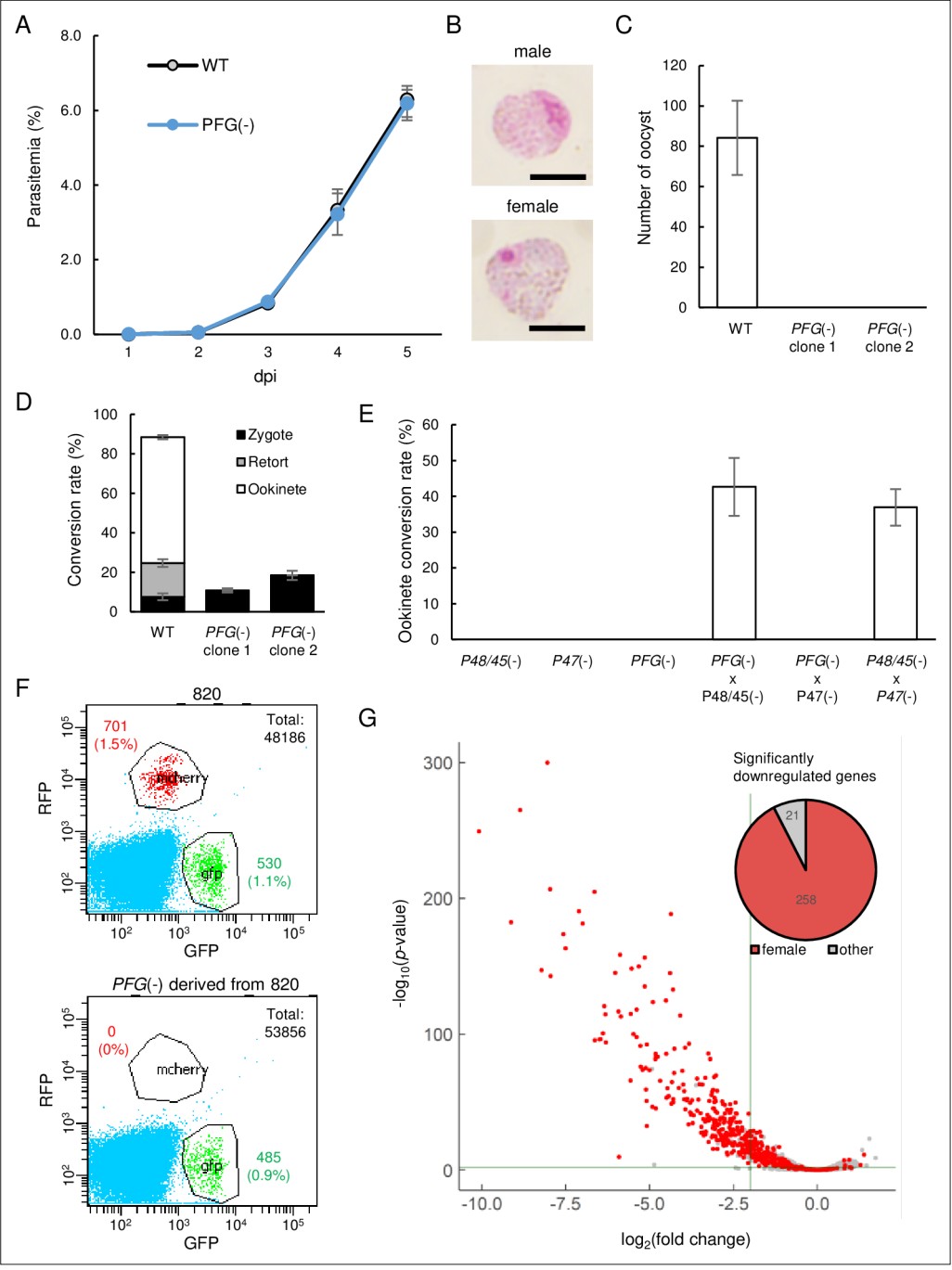

**Figure 2.** Development of female gametocytes was impaired in *PFG*(-)parasites. (**A**) Proliferation of blood-stage parasites in wild-type and *PFG*(−) parasites. Parasitemia was calculated by counting infected red blood cells using Giemsa staining of blood smears. Three biologically independent experiments were performed. Data are means ± SEM. (**B**) Giemsa-stained images of mature male and female gametocytes in *PFG*(-) parasites. Scale bars: 5 µm. (**C**) Number of oocysts in the mosquito midgut 14 d after infective blood meal of wild-type and *PFG*(-) parasites. Twenty mosquitoes were used to determine oocyst number. Data are means ± SEM. (**D**) Ookinete cultures of wild-type and *PFG*(-) parasites. Ratios of zygotes, retort-form, and banana-shaped ookinetes to all female-derived cells are shown in black, gray, and white, respectively. Data are means ± SEM from three biologically independent experiments. (**E**) A cross-fertilization experiment was performed between *PFG*(-), *P48/45*(-) (males are infertile), and *P47*(-) (females are infertile) parasites. The rate of conversion of females to mature ookinetes is shown. Data are means ± SEM from three biologically independent experiments. (**F**) Flow cytometry analysis of 820 parasites and *PFG*(-) parasites derived from 820 parasites. (**G**) Volcano plot showing differential expression of genes in *PFG*(-)

*Figure 2 continued on next page*

*Figure 2 continued*

compared to wild-type parasites. Red dots represent female-enriched genes, and horizontal and vertical lines indicate p-value of 0.001 and log₂(fold change) of –2, respectively. A pie graph on the top right of the plot area shows the number of female-enriched genes among the significantly downregulated genes.

The online version of this article includes the following source data and figure supplement(s) for figure 2:

**Figure supplement 1.** Preparation of PFG(-) and PFG(-)[820] parasites.

**Figure supplement 1—source data 1.** Original gel image for *Figure 2—figure supplement 1*.

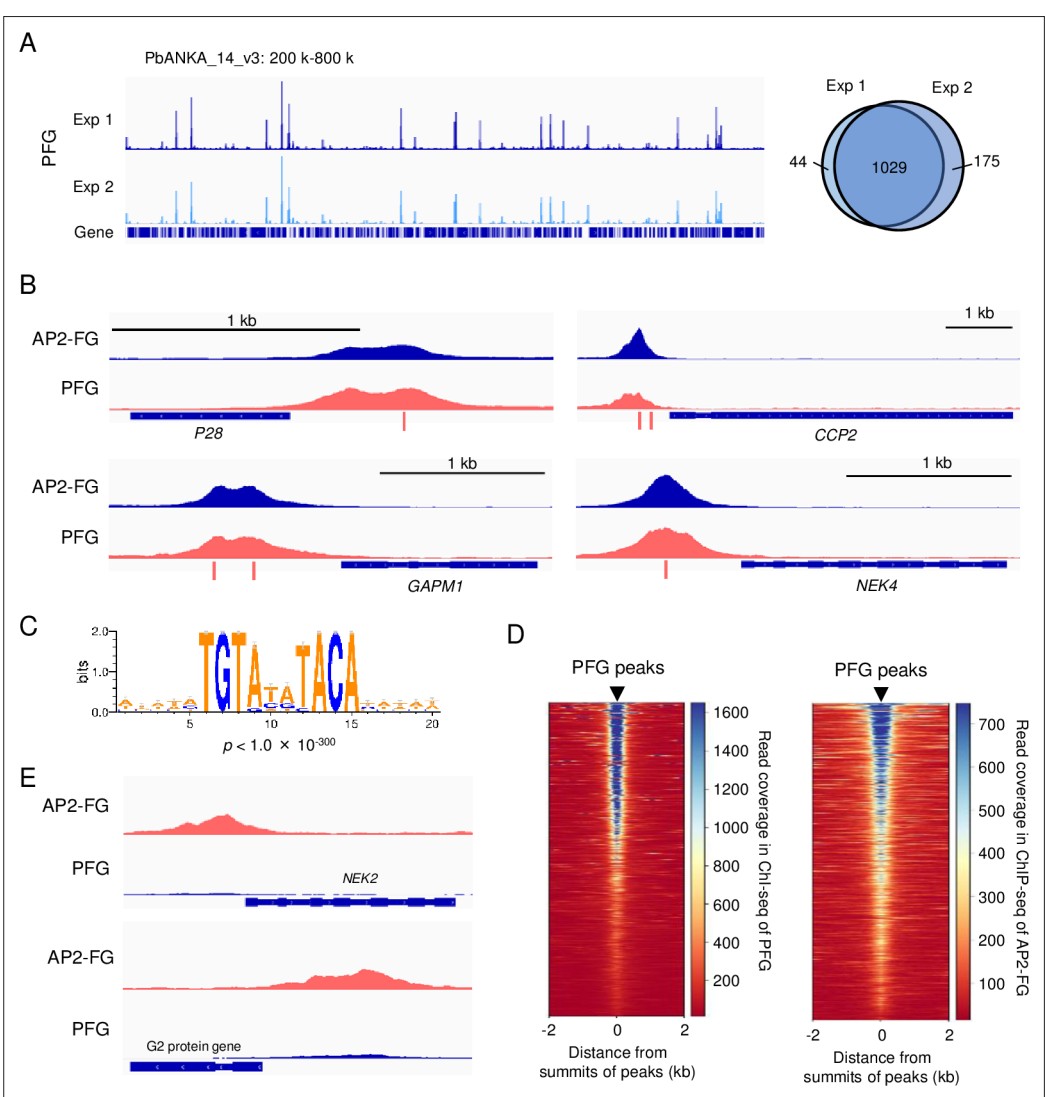

**Figure 3.** PFG is co-localized with AP2-FG. (**A**) IGV images from ChIP-seq experiments 1 and 2 of PFG on chromosome 14. Histograms show row read coverage in the ChIP data at each base. The Venn diagram on the right shows the number of peaks common in ChIP-seq experiments 1 and 2. Peaks were regarded as common when their summits were within 150 bp. (**B**) Representative peak images of PFG and AP2-FG upstream of genes that were significantly downregulated in *PFG*(-) parasites. Positions of ten-base motifs are indicated by red bars. ChIP-seq data for AP2-FG were obtained from a previous paper (*Yuda et al., 2019*). (**C**) The ten-base motif was enriched around the PFG peaks. Sequence logos were constructed using WebLogo 3 (http://weblogo. threeplusone.com/create.cgi). (**D**) Heat maps showing coverage in ChIP-seq of PFG (left) and AP-FG (right) with positioning summits of PFG peaks at the center. (**E**) IGV images of ChIP-seq peaks of AP2-FG that lacked the corresponding PFG peak. Histograms show the row read coverage in the ChIP data at each base.

those identified in experiment 2 (*Figure 3A*). These peaks were observed upstream of the genes significantly downregulated in *PFG*(-) parasites, suggesting that PFG is directly involved in the transcriptional activation of these genes (examples of graphic images are shown in *Figure 3B*). Statistical analysis of genomic sequences around the summits of peaks (common peaks, the same hereafter) showed that binding of PFG to the genome was associated with ten-base motif sequences, TGTRN-NYACA (*Figure 3C*), the female-specific *cis*-acting element identified in the ChIP-seq of AP2-FG (*Yuda et al., 2019*). In comparison with the graphical views, the peaks identified in the ChIP-seq of PFG were co-localized with those of AP2-FG (*Figure 3B*). An heat map positioned at the peak summit of PFG in the center showed that the position of the peaks was consistent with the ChIP-seq peaks of AP2-FG throughout the genome (*Figure 3D*). These results strongly suggest that these two TFs form a complex on the ten-base motif and cooperatively activate their targets. On the other hand, graphic images also showed that some of the AP2-FG peaks lacked the corresponding ChIP-seq peaks of PFG (*Figure 3E*).

## PFG is essential for AP2-FG binding to ten-base motif

Some sequence-specific TFs form heterodimeric complexes that bind specific DNA sequences. To examine whether this was the case with PFG and AP2-FG, ChIP-seq analyses for each of these two TFs were performed using parasites in which the other was disrupted.

ChIP-seq of PFG in *PFG::GFP* parasites with the AP2-FG gene disrupted (*PFG::GFP$^{AP2-FG(-)}$*), 1077 and 881 peaks were obtained, and 847 peaks were common between them (*Figure 4A* and *Supplementary file 2c and d*). These peaks showed no apparent changes compared to those obtained for the original *PFG::GFP* parasites (*Figure 4B*). The heat map showed that the positions of PFG peaks obtained in *PFG::GFP$^{AP2-FG(-)}$* parasites (common peaks, the same hereafter) were consistent with those obtained for *PFG::GFP* parasites throughout the genome, and vice versa (*Figure 4C*). Statistical analysis showed that the same ten-base motif obtained in the *PFG::GFP* parasites was enriched around the summits of the PFG peak obtained in *PFG::GFP$^{AP2-FG(-)}$* parasites (*Figure 4D*). These results indicate that PFG binds to the ten-base motif even in the absence of AP2-FG.

ChIP-seq of AP2-FG obtained in *AP2-FG::GFP* parasites with the PFG gene disrupted (*AP2-FG::GFP$^{PFG(-)}$*), 575 and 492 peaks were obtained, and 457 peaks were common between them (*Figure 4E* and *Supplementary file 2e and f*). The graphical appearance of the AP2-FG peaks changed drastically from that of the AP2-FG peaks obtained for the AP2-FG::GFP parasites. Comparison with PFG peaks showed that upon disruption of *PFG*, AP2-FG peaks common with PFG peaks disappeared, and only AP2-FG peaks not common with PFG peaks remained (*Figure 4F*). In line with this, statistical analysis showed that the ten-base motif disappeared and instead two distinct motifs, GCTCA and TGCACA, became the most enriched motifs around the summits of these peaks (common peaks, the same hereafter) with a p-value of $2.1 \times 10^{-27}$ and $1.3 \times 10^{-28}$, respectively (*Figure 4G*). These results suggest that PFG is essential for binding AP2-FG to the ten-base motif, that is, PFG mediates AP-FG binding to the ten-base motif. The results also suggest that AP2-FG binds to the genome in two distinct forms, with and without PFG, and that the former form binds to the ten-base motif, TGTRNNYACA, and the latter binds to other motifs, supposedly GCTCA or TGCACA. In the following, we refer to the form with PFG as cAP2-FG or the complex form, and the form without PFG as sAP2-FG or the single form.

## Ten-base motif is essential for binding of PFG to the genome

ChIP-seq of PFG in *PFG::GFP$^{AP2-FG(-)}$* parasites showed that PFG binds to the ten-base motif in the absence of AP2-FG. To demonstrate that the ten-base motif is essential for binding PFG to the genome, we performed a ChIP-qPCR assay using transgenic parasites in which the motif upstream of a target was mutated. A CPW-WPC family protein gene (PBANKA_1346300) (*Rao et al., 2016*) harbors a ChIP-seq peak for PFG and a ten-base motif under the summit of the peak. Three-point mutations were introduced into the motif using the CRISPR/Cas9 system, and *GFP* was fused to the PFG gene in these parasites (*Figure 4H*). Introducing these mutations reduced the %input value to background levels (*Figure 4I*), demonstrating that the motif is essential for PFG binding to the genome. At present, we speculate that PFG directly interacts with genomic DNA through two highly conserved regions; region 1 and region 2. However, these regions are not similar to any DNA binding domains reported thus far. In other apicomplexan orthologs, these two domains are located adjacent

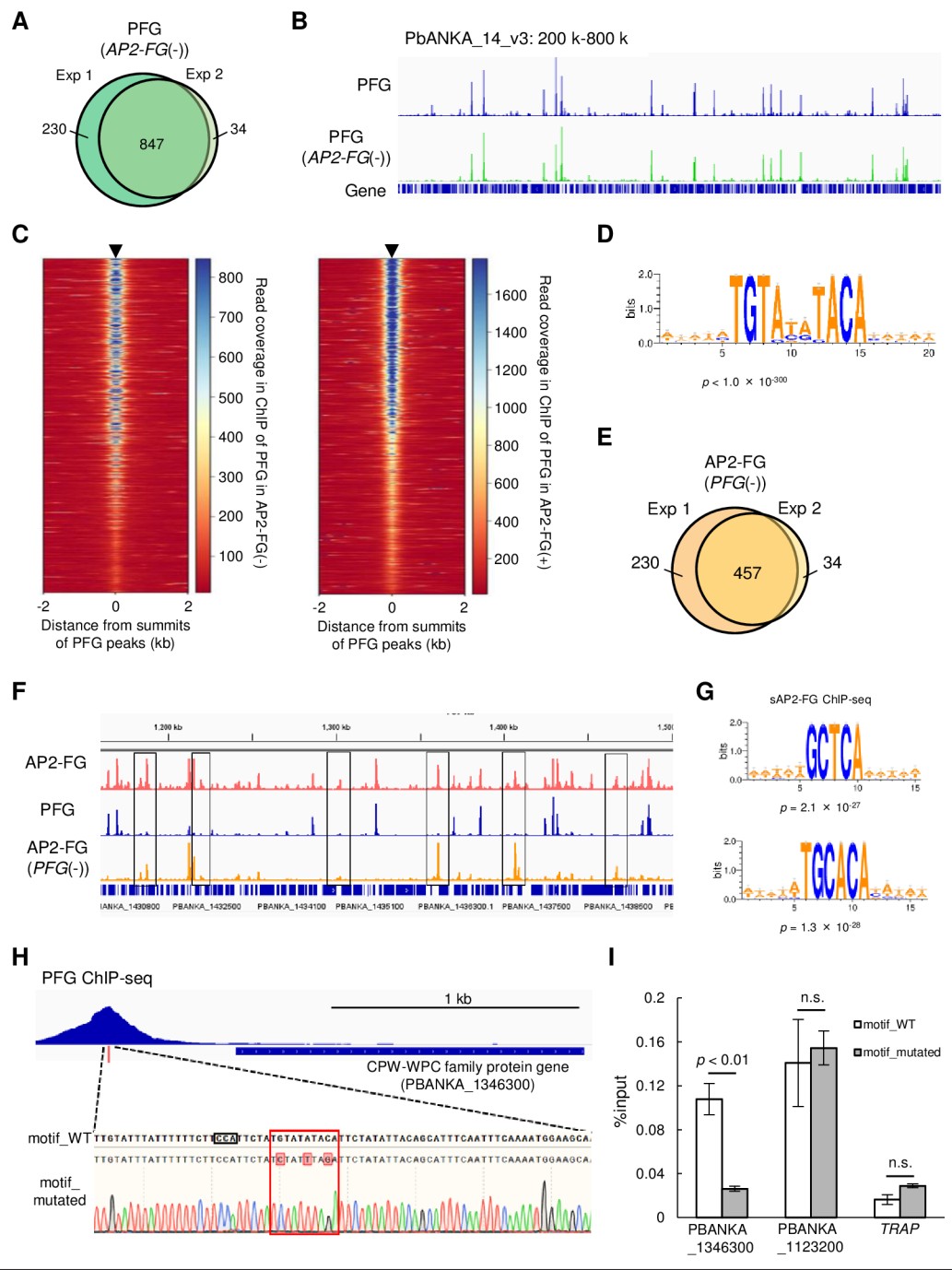

**Figure 4.** PFG mediates AP2-FG binding to the ten-base motif. (**A**) Venn diagram showing the number of common peaks between experiments 1 and 2 in ChIP-seq of PFG using *PFG::GFP*$^{AP2-FG(-)}$ parasites. Peaks were regarded as common when their summits were within 150 bp. (**B**) IGV images from the ChIP-seq peaks of PFG on part of chromosome 14. The top panel shows the ChIP peaks using *PFG::GFP* parasites, and the bottom panel shows those using *PFG::GFP*$^{AP2-FG(-)}$ parasites. Histograms show row read coverage in the ChIP data at each base. (**C**) Heat maps showing coverage in ChIP-seq of PFG using *PFG::GFP* parasites (left) and PFG using *PFG::GFP*$^{AP2-FG(-)}$ parasites, with positioning summits of peaks obtained in the other ChIP-seq at the center. (**D**) Motifs enriched around the summits of peaks identified in ChIP-seq of PFG using *PFG::GFP*$^{AP2-FG(-)}$ parasites. The sequence logo was depicted using WebLogo 3. (**E**) Venn diagram showing the number of common peaks between experiments 1 and 2 in ChIP-seq of AP2-FG using *AP2-FG::GFP*$^{PFG(-)}$ parasites. Peaks were regarded as common when their summits were within 150 bp. (**F**) IGV images of ChIP-seq peaks of AP2-FG and PFG in a region on which several AP2-FG peaks lack the corresponding PFG peak (chromosome 14:1,150–1500 kb). The top panel shows ChIP peaks

*Figure 4 continued on next page*

*Figure 4 continued*

of AP2-FG using *AP2-FG::GFP*. The middle and bottom panels show the ChIP-seq peaks of PFG using *PFG::GFP* and the ChIP peaks of AP2-FG using *AP2-FG::GFP*[PFG(-)] parasites, respectively. The AP2-FG peaks lacking their corresponding PFG peaks are highlighted by rectangles. Histograms show row read coverage in the ChIP data at each base. (**G**) Motifs enriched around the summits of peaks identified in the ChIP-seq of AP2-FG using *AP2-FG::GFP*[PFG(-)] parasites. Sequence logos are depicted using WebLogo 3. (**H**) Peak image of PFG and the ten-base motif under the summit upstream of a CPW-WPC family protein gene (PBANKA_1346300). The motif was mutated using the CRISPR/Cas9 system, and the mutation was confirmed by Sanger sequencing (the lowest panel; mutated nucleic acid residues are highlighted). The protospacer adjacent motif (PAM) sequence used for targeting is also highlighted by a rectangle. (**I**) ChIP-qPCR analysis of PFG in the upstream region of PBANKA_1346300. Gray and white bars indicate the results using wild-type and motif-mutated parasites, respectively. Three independent biological experiments were performed. The results are shown as a percentage input. Error bars indicate standard error. Experiments with another CPW-WPC family protein gene (PBANKA_1123200) and *TRAP* were performed as positive and negative controls, respectively.

The online version of this article includes the following source data and figure supplement(s) for figure 4:

**Figure supplement 1.** Preparation of PFG::GFP[AP2-FG(-)] parasites.

**Figure supplement 1—source data 1.** Original gel image for *Figure 4—figure supplement 1*.

**Figure supplement 2.** Preparation of *AP2-FG::GFP*[PFG(-)] parasites.

**Figure supplement 2—source data 1.** Original gel image for *Figure 4—figure supplement 2*.

---

to one another in the protein (*Figure 1A*). Therefore, these two regions may be separated by a long interval region but constitute a DNA binding domain of PFG as a result of protein folding.

## AP2-FG binds to the five-base motifs directly with its AP2 domain

The results of the ChIP-seq analysis of AP2-FG in *AP2-FG::GFP*[PFG(-)] parasites suggested that AP2-FG can bind to the genome directly through its AP2 domain. Statistical analysis showed that two distinct motifs, GCTCA and TGCACA, were enriched around the summits of ChIP-seq peaks (*Figure 4G*). To investigate whether the AP2 domain of AP2-FG binds to these sequences, we performed DNA immunoprecipitation followed by high-throughput sequencing (DIP-seq) using the recombinant AP2 domain of *P. berghei* AP2-FG and *P. berghei* genomic DNA (*Supplementary file 2g*). Around the summits of the peaks obtained by DIP-seq analysis, the five-base motif sequence GCTCA, identical to one of motifs enriched around the summits of ChIP-seq peaks described above, was highly enriched with a p-value of $2.6 \times 10^{-246}$ (*Figure 5A*). In addition, two motifs, GATCA and ACTCA, which may be variants of the GCTCA motif, were also enriched, with p-value of $2.0 \times 10^{-153}$ and $6.8 \times 10^{-51}$, respectively (*Figure 5A*). In contrast, the six-base motif, TGCACA, was not enriched around DIP-seq peaks (p-value=0.051), suggesting that it is not a binding motif of AP2-FG. Two minor motifs, GATCA and ACTCA, were also enriched around the summits of ChIP-seq peaks of sAP2-FG, with p-value of $2.0 \times 10^{-53}$ and $1.0 \times 10^{-13}$, respectively. In the ChIP-seq of sAP2-FG, 84.2% of the peaks contained at least one of these five-base motifs within 300 bp from the summits, and the average distance from the summits was 62.4 bp (*Figure 5B*). The most enriched motif matched well with the binding sequence of the AP2 domain of *P. falciparum* AP2-FG, which was reported by *Campbell et al., 2010*. Collectively, these results suggested that AP2-FG binds directly to these motifs through its AP2 domain.

A ChIP-qPCR assay was performed to confirm that this motif is essential for the binding of sAP2-FG to the upstream of target genes. NEK2 (PBANKA_1240700) is a Nima-related protein kinase essential for parasite transmission to mosquito vectors (*Reininger et al., 2009*). The NEK2 gene is one of the target genes of sAP2-FG, harboring one five-base motif sequence under the ChIP-seq peak. We prepared parasites with this motif mutated from *AP2-FG::GFP*[PFG(-)] parasites (*Figure 5C*) and performed a qPCR assay with anti-GFP antibodies. Upon addition of the mutation, the AP2-FG associated with the locus decreased to background levels (*Figure 5D*). These results demonstrated that AP2-FG binds to the five-base motif through its AP2 domain.

## Five-base motif acts as a *cis*-activating element on the promoter

The function of the five-a base motif as *cis*-acting element in the female-specific promoter was determined by promoter assays using the centromere plasmid (*Iwanaga et al., 2010*). The upstream region of the female-specific gene *P28* was used in this assay (*Figure 5E*). The region harbors one ten-base

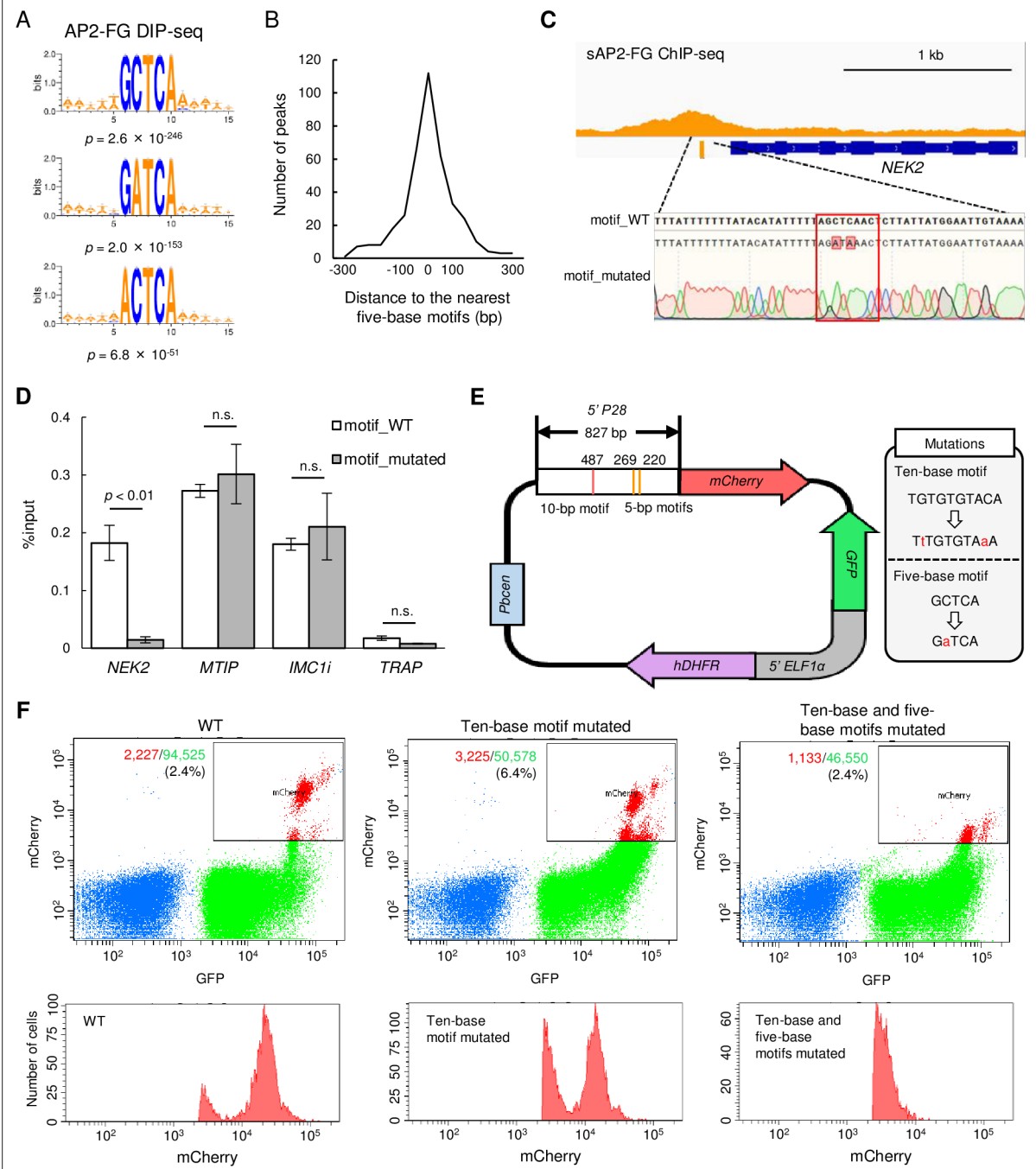

**Figure 5.** sAP2-FG binds to five-base motifs that function as a *cis*-acting element independent of the ten-base motif. (**A**) Motifs enriched around peaks identified in DIP-seq of the AP2 domain of AP2-FG. Sequence logos are depicted using WebLogo 3. (**B**) Histogram showing the distance between the peak summit and the nearest five-base motifs in (**A**) for each ChIP peak of sAP2-FG. (**C**) Peak image of sAP2-FG upstream of *NEK2* and the five-base motif under the summit. The motif was mutated, and the sequence was confirmed by Sanger sequencing (lowest panel). The protospacer adjacent motif (PAM) sequence used for targeting is highlighted by a rectangle. Mutated nucleic acid residues have also been highlighted. (**D**) ChIP-qPCR analysis of sAP2-FG in the upstream region of *NEK2*. Gray and white bars indicate the results obtained for wild-type and motif-mutated parasites, respectively. The results are shown as mean %input values from three biologically independent experiments. Error bars indicate standard error. Experiments with *MTIP* and *IMC1i* were performed as positive controls, and that in *TRAP* as a negative control. Statistical significance was determined using paired Student's *t*-test. (**E**) Schematic diagram of a *Plasmodium* centromere plasmid used to assessing promoter activity of *P28*. Mutations introduced into the ten-base and five-base motifs of the *P28* promoter are described in a gray box. Pbcen, a sequence of the *P. berghei* chromosome 5 centromere, 5'-ELF1α, a bidirectional promoter of the elongation 1α gene for conferring constitutive expression. (**F**) FACS analysis of parasites harboring the *P28*-reporter plasmid (**E**). Parasites were gated on forward-scatter and staining with Hoechst 33342, and then on GFP. The percentages of mCherry-positive parasites in all gated cells are shown on the left panel. The histogram on the right shows the number of mCherry-positive parasites at different signal strengths.

motif and three five-base motifs. The mCherry gene on the centromere plasmid was expressed under the control of this promoter region, and mCherry expression was analyzed by FACS (*Figure 5F*). Promoter activity was reduced by introducing mutations to the ten-base motifs, but the signals were still retained. The signals became almost undetectable by introducing mutations into the five-base motifs. These results demonstrate that the five-base motifs are female-specific *cis*-acting elements and function independently of the ten-base motif in the promoter.

## Genome-wide identification of cAP2-FG and sAP2-FG targets

Our ChIP-seq analyses suggested that the targets of AP2-FG were constituted by targets of cAP2-FG and sAP2-FG. To investigate which genes were commonly or uniquely regulated by these two forms of AP2-FG, we predicted the targets of cAP2-FG and sAP2-FG separately. As targets of cAP2-FG, 810 genes were predicted from ChIP-seq peaks of PFG obtained from *PFG::GFP* parasites (common peaks of two experiments, the same hereafter) (*Figure 6A* and *Supplementary file 3a*). These targets broadly contained genes related to female gametocyte-specific functions, including genes for female gametocyte development, such as genes for egress from the erythrocyte and fertilization, and genes for zygote/ookinete development, such as meiosis-related genes, genes for pellicular and subpellicular structures, ookinete surface or secretory protein genes, and crystalloid protein genes (*Supplementary file 3a*). As target genes for sAP2-FG, 321 were predicted by ChIP-seq in *AP2-FG::GFP*$^{PFG(-)}$ parasites (*Figure 6B* and *Supplementary file 3b*). Targets of sAP2-FG also contained many genes for female gametocyte-specific functions, but in smaller numbers than those in cAP2-FG targets. As a result of these target predictions, 967 genes were identified as targets for either cAP2-FG or sAP2-FG, and 164 genes were common between them (*Figure 6C*). Based on this classification, the cAP2-FG and sAP2-FG targets had the following features: first, genes known to be involved in gametocyte development are target genes of sAP2-FG, and most of them are also targets of cAP2-FG (thus belonging to common targets). Second, genes for zygote/ookinete development are targets of cAP2-FG, and some are also targets of sAP2-FG. Among these, most meiosis-related genes and crystalloid protein genes are the target genes unique to cAP2-FG.

Target prediction also showed that *PFG* is a target of sAP2-FG and cAP2-FG. Upstream of *PFG* harbors the binding sites of three different TFs: AP2-G, sAP2-FG (AP2-FG), and cAP2-FG (PFG) (*Figure 6D*). These results suggest that PFG is activated in early females by sAP2-FG and then by cAP2-FG in later female gametocytes. Meanwhile, the zygote TF AP2-Z was predicted to be a target of cAP2-FG only (*Nishi et al., 2022*). Together with the results obtained in previous studies (*Nishi et al., 2022*; *Yuda et al., 2019*), a cascade of TFs starting from AP2-G could be summarized as in *Figure 6E*.

## Impact of *PFG* disruption on the expression of AP2-FG target genes

To examine the independence of transcriptional activation by cAP2-FG from that of sAP2-FG, the impact of *PFG* disruption on the expression of AP2-FG target genes was assessed separately in the three groups described in *Figure 6C*, that is, unique for sAP2-FG, unique for PFG, and common for both, using RNA-seq data already presented in *Figure 2G*. Among the 279 genes significantly downregulated in *PFG*(-) parasites, 165 genes were targets for PFG (unique for PFG or common for sAP2-FG and PFG), and only 4 genes were targets unique to sAP2-FG. The $\log_2$ distribution (fold change = *PFG*(-)/wild type) in the three groups of target genes showed that the average value was significantly higher (i.e., less downregulated) in targets unique to sAP2-FG than in the other two groups (targets unique to cAP2-FG or common targets for both), with p-values of $1.3 \times 10^{-10}$ and $1.4 \times 10^{-5}$, respectively, by two-tailed Student's *t*-test (*Figure 6F*). In addition, the average $\log_2$ (fold change) value of the common target genes was relatively higher (i.e., less downregulated) than that of targets unique to PFG, suggesting that transcriptional activation by sAP2-FG partly complements the impact of PFG disruption on these common target. These results suggest that the trans-activity of cAP2-FG and sAP2-FG is independent of each other.

## Discussion

Genes transcribed in female gametocytes are involved in various steps of parasite transmission to the mosquito vector, including sex-specific differentiation in the vertebrate host and zygote/ookinete

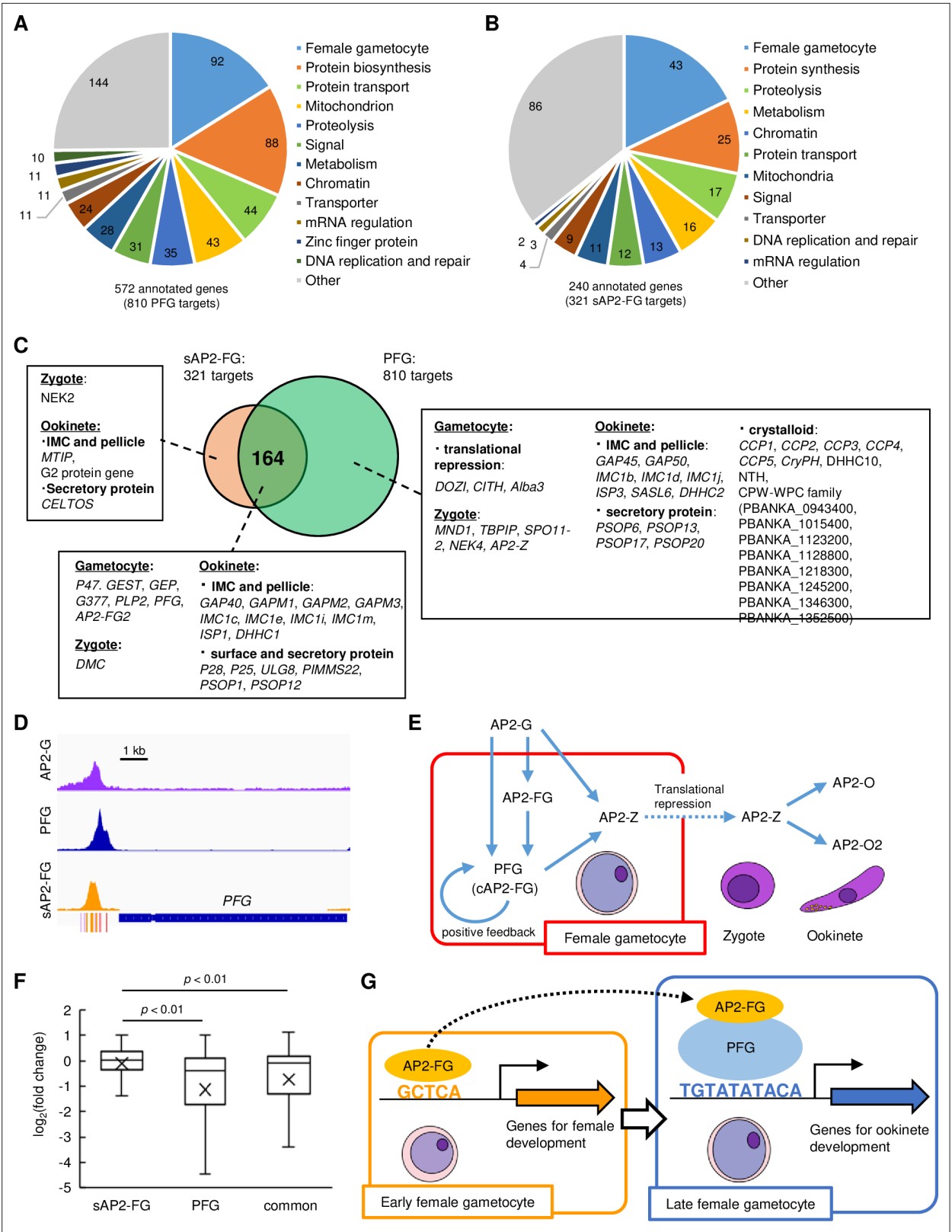

**Figure 6.** Target genes of PFG (cAP2-FG) and sAP2-FG contain different sets of genes. (**A**) Functional classification of the PFG target genes (cAP2-FG) (572 annotated genes in total). In this graph, different subgroups related to female gametocyte-specific functions are collectively shown as a group 'female-specific functions' (see also ***Supplementary file 3a***). Hypothetical protein genes were not included. The number of members in each group is shown in the chart. (**B**) Functional classification of the target genes of sAP2-FG (240 annotated genes in total). Hypothetical protein genes were

*Figure 6 continued on next page*

*Figure 6 continued*

not included. (**C**) Venn diagram showing an overlap between the target genes of sAP2-FG and PFG (cAP2-FG). The target genes in 'female-specific functions' are written according to their belongings: unique to sAP2-FG, common, and unique to cAP2-FG. The assignment of these genes into each subgroup was based on functional annotation in *Plasmo*DB and the following references (*Rao et al., 2016*; *Reininger et al., 2009*; *Nishi et al., 2022*; *Tremp et al., 2013*; *Bergman et al., 2003*; *Kariu et al., 2006*; *Andreadaki et al., 2020*; *Talman et al., 2011*; *Olivieri et al., 2015*; *Nishi et al., 2023*; *Ecker et al., 2008*; *Ukegbu et al., 2021*; *Sansam and Pezza, 2015*; *Pezza et al., 2007*; *Jenwithisuk et al., 2018*; *Kangwanrangsan et al., 2013*; *Dessens et al., 2011*; *Tremp et al., 2020*; *Santos et al., 2016*; *Santos et al., 2015*; *Wetzel et al., 2015*; *Saeed et al., 2020*). (**D**) IGV images from the ChIP-seq data of AP2-G, PFG, and sAP2-FG in the upstream region of *PFG*. The data of the ChIP-seq peaks for AP2-G were obtained from a previous study (*Yuda et al., 2021*). The peak region's binding motifs of AP2-G, ten-base and five-base motifs are indicated by purple, red, and orange bars. (**E**) A putative cascade of transcription factors (TFs) starting from AP2-G is suggested in this and previous studies. (**F**) Box-and-whisker plots showing $\log_2$(fold change) for target genes unique to sAP2-FG and PFG (cAP2-FG) and common for both. Cross marks in the boxes indicate the average values. Statistical significance was determined using paired Student's *t*-test. (**G**) A model of transcriptional regulation during female development. Female-specific genes harbor either or both of the female-specific *cis*-activating elements, five-base and ten-base elements in the upstream. In early female gametocytes, AP2-FG binds to a five-base *cis*-activating element via its AP2 domain and activates genes for gametocyte development and some genes for zygote/ookinete development. In the later stage, when PFG is highly expressed, AP2-FG is predominantly recruited to the PFG on the ten-base *cis*-activating element, and the AP2-FG/PFG complex activates a full repertoire of genes for zygote/ookinete development.

development that take place in the mosquito vector. In a previous study, we reported that this wide range of gene expression repertoires in females is regulated directly by a female-specific TF AP2-FG that binds to a ten-base *cis*-acting element upstream of these genes (*Yuda et al., 2019*). In this study, we report two findings that change our previous view. First, the binding of AP2-FG to the ten-base motif is mediated by another female-specific TF, PFG. Second, there is another *cis*-acting element in female-specific promoters, and AP2-FG binds to this element directly as a *trans*-activating factor. These findings suggest that transcriptional regulation of female-specific genes is not as simple as previously thought but is regulated by various promoters composed of different combinations of the two *cis*-acting elements. Considering the different expression profiles of PFG and AP2-FG, it is assumed that different expression profiles are generated among female-specific genes by the combination of these two *cis*-acting elements.

This study suggests that PFG is upregulated in two steps in female gametocytes; first by sAP2-FG, and then by cAP2-FG. Transcriptional activation of *PFG* by cAP2-FG constitutes a transcriptional positive feedback loop, suggesting that after activation by sAP2-FG, PFG maintains its transcription through this autoactivation mechanism. As the expression of PFG increases via this mechanism, AP2-FG recruited by PFG (cAP2-FG) increases and eventually becomes predominant in the transcriptional regulation of female gametocytes. *Figure 6G* illustrates this model. Considering that targetome of cAP2-FG contains many ookinete genes, transcripts for ookinete development would gradually increase in transcriptome of females. In this model, it is hypothesized that binding of AP2-FG contributes to the trans-activity of cAP2-FG. This is supported by the observation that expression of many unique targets of PFG decreases in *AP2-FG*-disrupted parasites (*Yuda et al., 2019*). However, it remains elusive how AP2-FG contributes to this activity. As discussed below, PFG may have its own trans-activity for the ten-base element. It is necessary to investigate the role of AP2-FG in cAP2-FG as a next step.

In *AP2-FG*-disrupted parasites, while female gametocytes display immature morphologies, a few develop into retort-form ookinetes. Because PFG is essential for the parasite's capacity to produce ookinetes, this phenotype suggests that PFG retains trans-activity in the absence of AP2-FG and partly promote ookinete development. The observation also supports this assumption that RFP signals in female gametocytes could still be detected in *AP2-FG*(-)[820] parasites by FACS (*Yuda et al., 2019*) even though RFP signals were reduced to undetectable level in *PFG*(-)[820] parasites under the same experimental conditions. This suggests that the promoter of the CCP2 gene, which is a target of PFG only, is still active in *AP2-FG*(-)[820] parasites. This putative trans-activity of PFG could also explain why the expression of *PFG* was still observed in *AP2-FG*-disrupted parasites. It is assumed that PFG continues to activate its transcription through a positive feedback loop once it is induced by AP2-G and complements the absence of AP2-FG. Based on this assumption and the model presented in *Figure 6G*, phenotypic differences between *PFG*-disrupted and *AP2-FG*-disrupted parasites can be explained as follows. *PFG*(-) parasites completely lose their ability to develop into ookinetes because the expression of genes necessary for ookinete development is significantly reduced. However, they manifest mature morphologies in females owing to the induction of the genes necessary for female

development by sAP2-FG. In contrast, *AP2-FG*(-) parasites show immature morphologies in females because the expression of genes involved in female development is downregulated in the early stage. Despite this immature morphology, they retain the ability to fertilize and develop into retort-form ookinetes because PFG is still expressed in *AP2-FG*(-) and activates genes for zygote/ookinete development.

*P. falciparum* gametocytes require 9–12 d to mature, which is much longer than that of *P. berghei* (*Gautret and Motard, 1999*). Meanwhile, it has been reported that the ten-base motif is highly enriched in the upstream regions of gametocyte-specific genes also in *P. falciparum* (*Young et al., 2005*). Thus, despite the difference in maturation periods, PFG is likely to play an important role in the transcriptional regulation of female gametocyte development in *P. falciparum*.

In conclusion, our results suggest that the two forms of AP2-FG, which correspond to distinct female-specific *cis*-acting elements, play pivotal roles in female gametocyte development and can generate variations among female-specific promoters. The present findings provide the possibility to predict the expression profile of each gene by analyzing its promoter sequence and further estimating its functions according to this regulation. In the next step, it will be necessary to compare the properties of the number of actual promoter activities in vivo and improve the model presented in this study. Such studies deepen our understanding of the sexual stage, including gametocyte development in erythrocytes, fertilization, meiosis, and development into ookinetes, and provide clues to interrupt parasite transmission from humans to mosquitoes.

## Materials and methods
### Parasite preparations

The ANKA strain of *P. berghei* was maintained in female BALB/c mice (6–10 wk of age). To examine the number of oocysts, infected mice were exposed to *Anopheles stephensi* mosquitoes. Fully engorged mosquitoes were selected and maintained at 20°C. The number of oocysts was evaluated 14 d after the infective blood meal. To prepare mature schizonts, infected mouse blood was cultured in the medium for 16 hr. Mature schizonts were purified using Nicoprep density gradient. To prepare gametocyte-enriched blood, mice were pretreated with hydrazine and infected with *P. berghei* by intra-abdominal injection of infected blood. After parasitemia increased to over 1%, they were treated with sulfadiazine for 2 d in drinking water (20 mg/L) to kill asexual parasites. After checking the exflagellation rate, whole blood was collected. All animal experiments were performed according to recommendations in the Guide for the Care and Use of Laboratory Animals of the National Institutes of Health in order to minimize animal suffering. All protocols were approved by the Animal Research Ethics Committee of Mie University (permit number 23-29).

### Preparation for transgenic parasites

Parasite transfection was performed as previously described (*Yuda et al., 2019*). Briefly, cultured mature schizonts were purified using a density gradient of Nicoprep, transfected with DNA constructs by electroporation, and injected intravenously into mice. These mice were treated with pyrimethamine to select the parasite integrated with the construct by double crossover homologous recombination. The parasites were cloned by limiting dilution to obtain transgenic parasite clones. To prepare the targeting construct, DNA fragments for recombination were annealed to each side of the DNA fragments containing *human DHFR* as a selectable marker gene by overlapping PCR. Constructs for parasites expressing the GFP-fused genes were prepared as follows. Briefly, DNA fragments corresponding to the 3′ portion of the gene were subcloned into the plasmid vector containing the GFP gene, the 3′-terminal portion of the *P. berghei* HSP70 gene, and a selectable marker cassette for expressing the *human DHFR* gene. The plasmid was separated from the vector backbone by digestion with restriction enzymes NotI and BanHI.

Transgenic parasites with mutated motif sequences were prepared using the CRISPR/Cas9 method, as reported previously (*Shinzawa et al., 2020*). Briefly, cultured mature merozoites of CAS9-expressing *P. berghei* parasites were transfected with a gRNA plasmid vector and a linear DNA template containing mutations in the motif sequences and injected intravenously into mice. These mice were treated with pyrimethamine for 2 d to deplete parasites in which the locus did not change.

The parasites were cloned by limiting dilution. Mutations in these motifs were confirmed by direct genome sequencing.

## Flow cytometric analysis

Flow cytometric analysis was performed using an LSR-II flow cytometer (BD Biosciences). In experiments using 820 parasites, the tail blood from infected mice was selected via gating with forward scatter and staining with Hoechst 33342 (excitation = 355 nm, emission = 450/50). The gated population was then analyzed for GFP fluorescence (excitation = 488 nm, emission = 530/30) and RFP fluorescence (excitation = 561 nm, emission = 610/20). In the promoter assay (using parasites transfected with a centromere plasmid), the tail blood from infected mice was selected via gating with forward scatter and staining with Hoechst 33342 (excitation = 355 nm, emission = 450/50), followed by GFP fluorescence (excitation = 488 nm, emission = 530/30). The gated population was analyzed for mCherry fluorescence (excitation = 561 nm, emission = 610/20). Analysis was performed using the DIVER program (BD Biosciences).

## ChIP-seq

Two infected mice were used in each ChIP-seq experiment. ChIP-seq experiments were performed as previously described (*Kaneko et al., 2015*). The gametocyte-enriched blood was filtered using a Plasmodipur filter to remove white blood cells and fixed in 1% paraformaldehyde for 30 min with swirling. Red blood cells were lysed in 0.84% NH$_4$Cl, and residual cells were lysed in a lysis buffer containing 1% SDS. The lysate was sonicated in Bioruptor 2. After centrifugation, the supernatant was diluted with dilution buffer, and chromatin was immunoprecipitated with Dynabeads protein A coated with an anti-GFP antibody (Abcam, ab290). DNA fragments were recovered from the precipitated chromatin and used for library preparation for sequencing. The library was prepared using a Hyper Prep Kit (KAPA Biosystems). Sequencing was performed on an Illumina NextSeq sequencer.

## Analysis of ChIP-seq data

Sequence data were mapped onto the *P. berghei* genome sequence (PlasmoDB, version 3) using BOWTIE2 software with default settings. Mapping data were analyzed using the MACS2 peak-calling algorithm. Conditions for peak calling were FDR < 0.01 and fold enrichment >2 or 3. To detect large peaks constituted from multiple peaks, the option 'Callsummit' was used. Mapped read data were visualized using IGV software. Sequences concentrated around the predicted summits of ChIP-seq peaks were investigated with Fisher's exact test carried out between 200 bp regions that had summits in the center and 200 bp regions excised from the genome, excluding the former regions, to cover the entire genome sequence. Sequences with the least p-values were combined with the common sequence motifs. Genes were determined as targets when their 1.2-kbp upstream regions harbored the predicted summits of ChIP-seq peaks. When the upstream intergenic region was less than 1.2 kbp, the entire intergenic region was used for target prediction. Two biologically independent ChIP-seq experiments were performed. The ChIP-seq data were deposited in the Gene Expression Omnibus (GEO) with the accession number GSE226028.

## DIP-seq

The recombinant AP2 domain of AP2-FG was prepared as a GST-fused protein using the same procedure as reported previously (*Yuda et al., 2021*). DIP was performed as reported previously (*Yuda et al., 2021*). Briefly, recombinant GST-fused AP2 domain of AP2-FG and genomic DNA fragments were incubated at room temperature for 30 min in 100 µL of binding/washing buffer. The mixture was then incubated with 50 µL glutathione sepharose resin (Cytiva) at room temperature with rotation. Resin was washed three times with 150 µL of binding/washing buffer, and bound protein was eluted with 10 mM glutathione solution. Recovered DNA (5 ng) was sequenced using the same procedures as in ChIP-seq. Input genomic DNA fragments were also sequenced as a control. Analysis of the sequence data was performed using the same procedures as in ChIP-seq. Data were deposited in the GEO database under accession number GSE226028.

## ChIP-qPCR

ChIP for ChIP-qPCR was performed using gametocyte-enriched mouse blood using the same procedures as in ChIP-seq. One infected mouse was used in each ChIP experiment. qPCR was performed using the TB Green Fast qPCR Mix (Takara Bio) according to the manufacturer's protocol. Three independent ChIP experiments were performed, and the percentage of input DNA (DNA extracted from lysate before ChIP) was compared between *P. berghei* parasites expressing GFP-fused TF and those with the mutated motifs. The primers used for qPCR are listed in *Supplementary file 4*.

## RNA-seq analysis

Gametocyte-enriched blood was passed through a filter and subjected to erythrocyte lysis in 0.84% $NH_4Cl$. According to the manufacturer's protocols, total RNA was extracted from residual cells using Isogen II (Nippon Gene). Libraries were prepared using the RNA Hyperprep kit (KAPA Biosystems) and sequencing was performed using an Illumina NextSeq sequencer.

## Analysis of RNA-seq data

Three independent experiments were performed using *PFG*-disrupted parasites. Read data were counted using FeatureCount software. RPKM (reads per kilobase of exon per million mapped reads) was calculated for each gene, and only genes with an RPKM maximum >20 in wild-type parasites were used for the following analyses. Additionally, genes located in the subtelomeric regions were excluded because of their variable expression among the clones. The ratio of read numbers and padj was calculated for each gene between wild-type and *gSNF2*-disrupted parasites using the DEseq2 software. Volcano plots of the obtained results were created using DDplot2 software. The RNA-seq data were deposited in GEO with the accession number GSE226028. The RNA-seq data deposited in GEO with accession number GSE198588 were used as data for the wild-type parasites.

## Acknowledgements

This work was supported by JSPS KAKENHI grant number 17H01542 to MY, as well as 23H02709 to MY, 21K06986 to TN, and 23K06515 to IK.

## Additional information

### Funding

| Funder | Grant reference number | Author |
| --- | --- | --- |
| Japan Society for the Promotion of Science | 23H02709 | Masao Yuda |
| Japan Society for the Promotion of Science | 21K06986 | Tsubasa Nishi |
| Japan Society for the Promotion of Science | 23K06515 | Izumi Kaneko |
| Japan Society for the Promotion of Science | 17H01542 | Masao Yuda |

The funders had no role in study design, data collection and interpretation, or the decision to submit the work for publication.

### Author contributions

Yuho Murata, Conceptualization, Investigation, Writing - original draft; Tsubasa Nishi, Investigation, Writing - review and editing; Izumi Kaneko, Investigation; Shiroh Iwanaga, Resources, Methodology; Masao Yuda, Conceptualization, Investigation, Writing - original draft, Project administration, Writing - review and editing

### Author ORCIDs

Masao Yuda http://orcid.org/0000-0002-3416-5132

## Ethics

All experiments in this study were performed following the recommendations in the Guide for the Care and Use of Laboratory Animals of the National Institutes of Health to minimize animal suffering and were approved by the Animal Research Ethics Committee of Mie University, Mie, Japan (permit number 23-29).

Reviewer #1 (Public Review): https://doi.org/10.7554/eLife.88317.3.sa1
Reviewer #2 (Public Review): https://doi.org/10.7554/eLife.88317.3.sa2
Reviewer #3 (Public Review): https://doi.org/10.7554/eLife.88317.3.sa3
Author Response https://doi.org/10.7554/eLife.88317.3.sa4

# Additional files

### Supplementary files

• Supplementary file 1. Differential expression analysis between wild-type and *PFG*(-) parasites.

• Supplementary file 2. Peaks identified in ChIP-seq experiments and their nearest genes. (a) Peaks identified in ChIP-seq of PFG with their nearest genes (experiment 1). (b) Peaks identified in ChIP-seq of PFG (experiment 2). (c) Peaks identified in ChIP-seq of PFG using *AP2-FG*(-) parasites (experiment 1). (d) Peaks identified in ChIP-seq of PFG using AP2-FG(-) parasites (experiment 2). (e) Peaks identified in ChIP-seq of sAP2-FG (experiment 1). (f) Peaks identified in ChIP-seq of sAP2-FG (experiment 2). (g) Peaks identified in DIP-seq using recombinant AP2 domain of AP2-FG and *P. berghei* genomic DNA.

• Supplementary file 3. Target genes predicted from ChIP-seq peaks. (a) PFG target genes predicted from ChIP-seq peaks. (b) sAP2-FG target genes predicted from ChIP-seq peaks.

• Supplementary file 4. List of primers used in this study.

• MDAR checklist

### Data availability

Sequencing data have been deposited in GEO under the accession code GSE226028.

The following dataset was generated:

| Author(s) | Year | Dataset title | Dataset URL | Database and Identifier |
|---|---|---|---|---|
| Yuda M | 2024 | PFG controls gene expression in female gametocytes cooperatively with AP2-FG | https://www.ncbi.nlm.nih.gov/geo/query/acc.cgi?acc=GSE226028 | NCBI Gene Expression Omnibus, GSE226028 |

The following previously published datasets were used:

| Author(s) | Year | Dataset title | Dataset URL | Database and Identifier |
|---|---|---|---|---|
| Yuda M | 2019 | Sex-specific gene regulation in malaria parasites by an AP2 Family Transcription Factor | https://www.ncbi.nlm.nih.gov/geo/query/acc.cgi?acc=GSE114096 | NCBI Gene Expression Omnibus, GSE114096 |
| Yuda M | 2022 | Differentiation of malaria male gametocytes is initiated by recruitment of a chromatin remodeler to male-specific cis-acting elements | https://www.ncbi.nlm.nih.gov/geo/query/acc.cgi?acc=GSE198588 | NCBI Gene Expression Omnibus, GSE198588 |

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
