## [Editor Report · eLife assessment]

This study offers **important** insights into the transcriptional regulatory networks driving female gametocyte maturation in rodent malaria parasites. The work is based on **solid** methodology and shows how two female-specific transcription factors, AP2-FG and PFG (aka Fd2), cooperate to upregulate the expression of genes required for development after fertilization occurs in the mosquito midgut. This study will be of interest to scientists working on sexual differentiation and gene regulation in *Plasmodium* and other apicomplexan parasites.

---

## [Referee Report · Reviewer #1 (Public Review)]

Gametocytes are erythrocytic sexual stages of the malaria-causing parasite Plasmodium, and are essential for parasite transmission and reproduction in the mosquito vector. In this study, Murata et al investigated the mechanisms of gene regulation in female gametocytes in the rodent malaria model parasite Plasmodium berghei. According to current views, gene regulation in Plasmodium parasites is dominated by the family of AP2 transcription factors (TFs), such as the AP2-G TF, which drives sexual commitment. The same authors previously identified one AP2 TF, called AP2-FG, as an essential TF mediating differentiation of female gametocytes. Here, they identified a novel protein, called PFG (for partner of AP2-FG, also described as Fd2 in a recently published study), which cooperates with AP2-FG to regulate a subset of female gametocyte genes.

PFG was identified among AP2-G targets, but possesses no known DNA binding or other characterized domain. The authors show that PFG-knockout P. berghei parasites can form male and female gametocytes yet cannot transmit to mosquitoes, due to a defect in female gametocyte development. Using RNA-seq, they show that many female-specific genes are down-regulated in PFG(-)parasites. Chromatin immunoprecipitation combined with DNA sequencing (ChIP-seq) revealed that PFG colocalizes with AP2-FG on a ten-base motif that is enriched upstream of female-specific genes. Importantly, the ChIP-seq profile of PFG is unchanged in the absence of AP2-FG, suggesting that PFG binds to DNA independently of AP2-FG. Mutation of the ten-base motif in one target gene using CRISPR-Cas9 demonstrates that this motif is required for PFG localization at the gene locus. The data also show that binding of AP2-FG is affected in the absence of PFG, with disruption of AP2-FG interaction with the ten-base motif, but conservation of AP2-FG binding to distinct five-base motifs. Using a recombinant AP2 domain from AP2-FG, the authors demonstrate that the AP2 domain of AP2-FG binds to the five-base motifs. Using CRISPR they show that disruption of the five-base motifs in the genome abrogates AP2-FG binding, and using a reporter expression system they confirm that these motifs act as a cis-activating promoter element.

Through the analysis of target genes based on the presence of the ten- versus five-base motifs, the authors propose a model where AP2-FG can function in two forms, associated or not with PFG, and acting on the ten- or five-base motifs, respectively, to regulate distinct gene subsets during development of female gametocyte development.

The paper is well written, with a clear narrative, and the work is very well performed, relying on robust molecular approaches. Generally the conclusions and the model proposed by the authors are well supported by the data. Nevertheless, the study as it stands raises a number of questions. While the data convincingly show that PFG and AP2-FG cooperate to regulate the expression of a subset of female-specific genes, the paper does not show whether the two proteins actually interact with each other to form a complex. Also, how PFG binds to DNA and whether the protein has transactivating activity remains elusive, as the protein apparently possesses no known DNA-binding or activating domain. These points could be discussed in more detail in the manuscript and/or be the subject of follow up studies.

In summary, this work reveals the essential role of a Plasmodium protein with no known DNA binding or regulatory domain, illustrating that unknown facets remain to be uncovered in this fascinating pathogen.

---

## [Referee Report · Reviewer #2 (Public Review)]

Murata et al have characterized a transcription activator previously identified in an earlier genetic screen by Russell et al (named Fd2; for female-defective 2), here named PFG. The authors show solid evidence that PFG is a partner of the previously described transcription factor AP2-FG and describe three sets of genes: genes activated by PFG or AP2-FG alone and genes activated by the complex. The authors also show differential binding to the target DNA sequences by AP2-FG to either a 10bp, if in a complex with PFG or a 5bp motif if alone. In all, this is a useful study which further elucidates the underlying regulatory network that drives development of sexual stages and ultimately transmission to mosquitoes. The data presented are clear and solid and the conclusions drawn are mostly supported by the results shown.

A few comments:

Given that the transcriptional programme is so dynamic, the timing of the ChIP-seq experiments is crucial. Could the authors clarify the timings of the different ChIP-seq experiments (AP2-FG, PFG, PFG in AP2-FG-, AP2-FG in PFG-, ...)

Fig 4c is an example of great overlap of peaks, but it would be helpful if the authors could quantify the overlaps between experiments (and describe the overlap parameters used).

It remains unclear if AP2-FG and PFG interact directly or if they bind sequentially in the transcriptional activation process. Perhaps they are part of a larger complex? Immunoprecipitation followed by mass spectrometry of the GFP-tagged version of PFG in the presence and absence of AP2-FG would be highly informative.

---

## [Referee Report · Reviewer #3 (Public Review)]

This study is well designed and executed and provides new and important insights into the role of two TFs during the maturation of female gametocytes and fertilization in the mosquito midgut. However, it but would benefit from a more thorough characterization of the phenotype to understand at which step of development these factors are required.

Overall the authors have shown only limited willingness to comprehensively address reviewer concerns and incorporate their suggestions.

---

## [Author Response]

The following is the authors’ response to the original reviews.

**Reviewer #1 (Recommendations For The Authors):**
The manuscript is very well written, the data are clearly presented and the methodology is robust. I only have suggestions to improve the manuscript, to make the study more appealing or to discuss in more detail some questions raised by the work.1. In the study as it stands, PFG seems to come out of the blue. The authors apparently selected this protein based on sequence conservation between species but this is unlikely to be sufficient to identify novel TFs. Explaining in more detail the reasoning that led to PFG would make the story more appealing. Perhaps PFG was identified through a large reverse genetics screening?

Response: Thank you for your suggestion. We identified this gene solely by the strategy we described in the manuscript. We decided on this strategy based on the findings of our previous study on AP2-Family TFs, whose DNA binding domains are highly conserved among Plasmodium orthologues. Using this screening strategy, we identified a novel AP2 family TF AP2-Z. The results of the present study demonstrated that this strategy is applicable to TFs other than those belonging to the AP2 family. We are aware that this strategy is not all-encompassing. In fact, we failed to identify HDP1 as a candidate TF when it was also in the target list of AP2-G. However, at present, this is our primary strategy for identifying novel TFs in the targetome.

1. The authors propose that PFG and AP2-FG form a complex, but this is actually not shown. Did they try to document a physical interaction between the two proteins, for example using co-IP?

Response: Even when the two molecules were identified to be at the same position by ChIPseq, it cannot be concluded that they form a physical complex because it is possible that they competitively occupy the region. However, in this study, we performed ChIP-seq in the absence of PFG and demonstrated that the cAP2-FG peaks disappeared while those of sAP2-FG remained. This result can only be explained by the two proteins forming a complex at this region, which excludes the possibility that AP2-FG binds the region independently.

1. It is unclear how PFG can bind to DNA in the absence of DNA-binding domain. Did the authors search for unconventional domains in the protein? This should be at least discussed in the manuscript.

Response: We speculate that the two highly conserved regions, region 1 and region 2, function as DNA-binding domains in PFG. However, this domain is not similar to any DNA binding domains reported thus far. A straightforward way to demonstrate this would be to perform in vitro binding assays using a recombinant protein. However, thus far, we have not succeeded in obtaining soluble recombinant proteins for these regions. We have added the following sentences to the results section.

“At present, we speculate that PFG directly interacts with genomic DNA through two highly conserved regions; region 1 and region 2. However, these regions are not similar to any DNA binding domains reported thus far. In other apicomplexan orthologues, these two domains are located adjacent to one another in the protein (Fig. 1A). Therefore, these two regions may be separated by a long interval region but constitute a DNA binding domain of PFG as a result of protein folding.”

1. How do the authors explain that PFG is still expressed in the absence of AP2-FG? Is AP2G alone sufficient to express sufficient levels of the protein? Is PFG down-regulated in the absence of AP2-FG?

Response: Our previous ChIP-seq data indicate that PFG is a target of AP2-G. According to the study by Kent et al. (2018), this gene is up-regulated in the early period following conditional AP2-G induction. The results of the present study showed that PFG is capable of autoactivation through a transcriptional positive feed-back loop. These results suggest that PFG can maintain its expression to a certain level once activated by AP2-G, even in the absence of AP2-FG. In our previous microarray analysis, significant decreases in PFG expression were not observed in AP2-FG-diaruptedparasites.

1. How do AP2-FG regulated genes (based on RNAseq) compare with the predicted cAP2FG/sAP2-FG predicted genes (based on ChIPseq)? Are the two subsets included in the genes that are actually down-regulated in AP2-FG(-)?

Response: Disruption of the AP2-FG gene impairs gametocyte development. We considered that the direct effect of this disruption would be difficult to analyze in gametocyte-enriched blood, in which gametocytes are pooled during sulfadiazine treatment to deplete asexual stages. Therefore, in our previous paper, we performed microarray analysis between WT and KO parasites to detect the direct effect of AP2-FG disruption on target gene expression, using mice which were synchronously infected with parasites. According to our results, 206 genes were down-regulated in AP2-FG-disrupted parasites. Of these genes, 40 and 117 were targets of sAP2-FG and cAP2-FG, respectively. However, it is still possible that a significant proportion of genes were indirectly down-regulated by AP2-FG disruption, which may impair gametocyte development. Moreover, based on the results of the present study, expression of a significant proportion of AP2-FG target genes could be complemented by PFG transcription. We believe that it would be difficult to compare the direct effects of these TFs on gene expression via transcriptome analysis (therefore, targetome analysis is important). In this study, we compared the expression of target genes of sAP2-FG and cAP2FG between PFG(-) and WT parasites. We expected that down-regulation of PFG (cAP2FG) targets would be complemented with transcription by sAP2-FG.

1. Minor points-Page 5 Line 10, remove "as"

Response: We have corrected this.

-Page 7 Lines 4-13: is it possible to perform the assay in PFG(-) parasites?

Response: Thank you for your question. Even when the marker gene expression was decreased in PFG(-) parasites, we cannot conclude the reason to be a direct effect of the mutation. To determine the function of the motif, it is necessary to perform the assay using wild-type parasites.

-Page 7 Line 45: Fig6C instead of 5C

Response: Thank you for pointing this out. We have corrected this.

-Page 8 Line 27: "decreases"

Response: Thank you for pointing this out. We have corrected this.

-Page 8 Line 36: PFG instead of PGP

Response: We have corrected this.

-Page 8 Line 39: remove "the fact"

Response: We have removed this word.

-Page 8 Line 42: Fig6G instead of 5G

Response: We have corrected this.

-Page 8 Line 43: PFG instead of PGP

Response: We have corrected this.

-Page 9 Line 23: "electroporation"

Response: We have corrected this.

-Page 9 Line 32: "BamHI"

Response: We have corrected this.

-Fig 2E: in the crosses did the authors check oocyst formation in the mosquito?

Response: We did not check oocyst formation because abnormalities in males may not affect oocyst formation.

-Page 17, legend Fig3, Line 14, there is probably an inversion between left and right for PFG versus AP2-FG (either in the legend or in the figure)

Response: Thank you for pointing this out. PFG peaks are located in the center in both heat maps. The description “AP2-FG peaks” over the arrowhead in the left map was incorrect. We have corrected this to “PFG peaks”. The peaks in the left heat map must be located in the center; thus, this figure might be redundant.

**Reviewer #2 (Recommendations for the Authors):**
Could the authors please state in the results section that PFG stands for partner of AP2FG.

Response: Thank you for the comment. We have added the following to the results section:

“Through this screening, a gene encoding a 2709 amino acid protein with two regions highly conserved among Plasmodium was identified (PBANKA0902300, designated as a partner of AP2-FG PFG; Fig. 1A).”

Given that the transcriptional program is so dynamic, the timing of the ChIP-seq experiments is crucial. Could the authors clarify the timings of the different ChIP-seq experiments (AP2-FG, PFG, PFG in AP2-FG-, AP2-FG in PFG-, ...)

Response: Thank you for the comment. To deplete any parasites in the asexual stages, all ChIP-seq experiments in this study were performed using blood from mice treated with sulfadiazine, namely, gametocyte-enriched blood. As the reviewer points out, timing is important, and samples from the period when TFs are maximally expressed are optimal for ChIP-seq. However, when parasites in the asexual stages are present, the background becomes higher. Thus we usually use gametocyte-enriched blood for ChIP-seq when expression of the TF is observed in mature gametocytes. The exception was our ChIP-seq analysis of AP2-G, because is not present in mature gametocytes.

Fig 4c is an example of great overlap of peaks, but it would be helpful if the authors could quantify the overlaps between experiments (and describe the overlap parameters used).

Response: According to the comment, we have created a Venn diagram of overlapping peaks (attached below). However, the peaks used for this Venn diagram were selected after peakcalling via fold-enrichment values. Thus, even if the counterpart of a peak is absent in these selected peaks (non-overlapping peaks in the Venn diagram), it does not indicate that it is absent in the original read map. We believe the overlap of peaks would be estimated more correctly in the heat maps.

**Author response image 1. sa4fig1:** Legged: The Venn diagram shows the number of common peaks between these ChIP seq experiments (distance of peak summits < 150).

Additionally, how were the promoter coordinates used for each gene when they associate ChIP peaks to a gene target. Did the authors choose 1-2kb? Or use a TSS/5utr dataset such as Adjalley 2016 or Chappell 2020?

Response: We selected a 1.2 Kbp region for target prediction based on our previous studies. As the reviewer pointed out, target prediction using TSS information may be more accurate. However, reliable TSS information is not available for P. berghei to the best of our knowledge.

The two papers are studies on *P. falciparum*.

In the absence of evidence of physical interaction, it remains unclear if AP2-FG and PFG actually interact directly or as part of the same complex. A more detailed characterisation with IPs/co-IPs followed by mass spectrometry of the GFP-tagged version of PFG in the presence and absence of AP2-FG would be highly informative.

Response: Thank you for the comment. Even when these two TFs occupy the same genomic region, it cannot be conclusively said that they exist at the same time in the region: they might competitively occupy the region. However, we showed that the cAP2-FG peaks disappear from the region when PFG was disrupted, while sAP2-FG peaks remain. We believe that this is evidence that the two TFs physically interact with each other.

It was not clear if the assessment of motif binding using cytometry was performed using all the required controls and compensation. This section should be clarified.

Response: Thank you for the comment. Condensation was performed using parasites expressing a single fluorescent protein. The results are attached below. The histogram of mCherry using control parasites expressing GFP under the control of the HSP70 promoter is also attached.

**Author response image 2. sa4fig2:** 

However, we found that descriptions of the filters for detecting red signals were not correct. This assay was performed using parasites which expressed GFP constitutively and mCherry under the control of the p28 promoter. These two fluorescent proteins were excited by independent lasers (488 and 561, respectively), and the emission spectra were detected using independent detectors (through 530/30 and 610/20 filters, respectively). We have revised the description regarding our FACS protocols as follows:

“Flow cytometric analysis was performed using an LSR-II flow cytometer (BD Biosciences). In experiments using 820 parasites, the tail blood from infected mice was selected via gating with forward scatter and staining with Hoechst 33342 (excitation = 355 nm, emission = 450/50). The gated population was then analyzed for GFP fluorescence (excitation = 488 nm, emission = 530/30) and RFP fluorescence (excitation = 561 nm, emission = 610/20). In the promoter assay (using parasites transfected with a centromere plasmid), the tail blood from infected mice was selected via gating with forward scatter and staining with Hoechst 33342 (excitation = 355 nm, emission = 450/50), followed by GFP fluorescence (excitation = 488 nm, emission = 530/30). The gated population was analyzed for mCherry fluorescence (excitation = 561 nm, emission = 610/20). Analysis was performed using the DIVER program (BD Biosciences).”

Minor points:Page 4, line 37: The authors should specify the timing of expression of AP2-FG on the text.

Response: We have added the following description to the text.

“The timing of the expression was approximately four hours later than that of AP2-FG, which started at 16 hpi (9).” .

Ref 9 and 17 are repeated

Response: Thank you for pointing this out. We have corrected this.

Fig 1D and 1F do not have scale bars

Response: We have added scale bars to Fig. 1D.

We have not changed Fig. 1F, because we believe that the scales can be estimated from the size of the erythrocyte.

Page 5, line 29-30. Could the authors specify how many and which of the de-regulated genes have a PFG in their promoter.

Response: Thank you for the comment, As described in a later section (page 7; Impact of PFG disruption on the expression of AP2-FG target genes), among the 279 genes significantly downregulated in PFG(-) parasites, 165 genes were targets for PFG (unique for PFG or common for sAP2-FG and PFG). In contrast, only four genes were targets unique to sAP2-FG. Therefore, 165 genes harbor the upstream peaks of PFG. These genes are shown in Table S1.

Fig 5F. in the methods associated with this figure there seems to be a mixup with the description of the lasers. In addition, given the spillover of the red and green signal between detectors this experiment needs compensation parameters. The authors should provide the gating strategy before and after compensation as this is critical for the correct calculation of the number of red parasites. Indeed, the lowest red cloud on the gate shown could be green signal spill over.

Response: Thank you for the comment. As described above, there were some incorrect descriptions about the conditions of our FACS protocols in the methods section. We have revised them.

-Page 7, line 19. Could the authors explicitly say in the text that the 810 genes are those with 1 (or more?) PFG peaks in their promoter (out of a total of 1029) to best guide the reader. Additionally, it is important to define the maximum distance allowed between a peak and CDS for it to be associated with said CDS.

Response: We have revised Table S2 by adding the nearest genes. The revised table shows the relationship between a PFG peak and its nearest genes, together with their distances.

Page 7, line 45: fig 6c, not 5c

Response: Thank you for the comment. We have corrected this.

Page 7 last paragraph: This section is very hard to follow. For instance, on line 50 do the authors mean that the sAP2-FG unique targets are LESS de-regulated? On line 51: do the authors mean unique targets of cAP2-FG or unique targets of PFG? Line 53: do the authors mean that genes expressed in the "common" category are LESS de-regulated than the PFG unique targets?

Response: We are sorry for the lack of clarity; after reviewing the manuscript, it appears to be unclear what the fold change means in this section. Here, fold change means the ratio of PFG(-)/wild type. Thus “High log2(fold change) value” means that the genes were less downregulated. We have revised the description as follows:

“The log2 distribution (fold change = PFG(-)/wild type) in the three groups of target genes showed that the average value was significantly higher (i.e., less down-regulated) in targets unique to sAP2-FG than in the other two groups (targets unique to cAP2-FG or common targets for both), with p-values of 1.3 × 10-10 and 1.4 × 10-5, respectively, by two-tailed Student’s t-test (Fig. 6F). In addition, the average log2 (fold change) value of the common target genes was relatively higher (i.e., less down-regulated) than that of targets unique to PFG, suggesting that transcriptional activation by sAP2-FG partly complements the impact of PFG disruption on these common targets.”

Page 8, line 42: Fig 6G, not 5G

Response: Thank you for pointing this out. We have corrected this.

**Reviewer #3 (Recommendations For The Authors):**
1. The gene at the center of this study (PBANKA_0902300) was identified in an earlier genetic screen by Russell et al. as being a female specific gene with essential role in transmission and named Fd2 (for female-defective 2). Since this name entered the literature first and is equally descriptive, the Fd2 name should be used instead of PFG to maintain clarity and avoid unnecessary confusion. Surprisingly, this study is neither cited nor acknowledged despite a preprint having been available since August of 2021. This should be remedied.

Response: Thank you for the comment. We have added the paper by Russell et al. accordingly and mentioned the name FD2 in the revised manuscript. However, we have retained the use of PFG throughout the paper. We believe that this usage of PFG shouldn’t be confusing, as FD2 has only been used in one previous paper. We have added the following:

“Through this screening, a gene encoding a 2709 amino acid protein with two regions highly conserved among Plasmodium was identified (PBANKA0902300, designated as a partner of AP2-FG PFG; Fig. 1A). This gene is one of the P. berghei genes that were previously identified as genes involved in female gametocyte development (named FD2), based on mass screening combined with single cell RNA-seq (ref).”

1. While it isn't really important how the authors came to arrive at studying the function of Fd2, the rationale/approach given in the first paragraph of the result section seems far too broad to lead to Fd2, given that it lacks identifiable domains and many other ortholog sets exist across these species.

Response: We selected this gene from the list of AP2-G targets as a candidate for a sequence-specific TF based on the hypothesis that the amino acid sequences of DNAbinding domains are highly conserved. We successfully identified two TFs (including PFG) using this method. However, there may be TFs that do not fit this hypothesis which are also targets of AP2-G. In fact, we were unable to identify HDP1 as a TF candidate, despite being a AP2-G target.

1. Fig. 1A-C: Gene IDs for the orthologs should be provided, as well as the methodology for generating the alignments.

Response; We have added the gene IDs and method for alignment in the legend as follows:

(A) Schematic diagram of PFG from P. berghei and its homologs in apicomplexan parasites. Regions homologous to Regions 1 and 2, which are highly conserved among Plasmodium species, are shown as yellow and blue rectangles, respectively. Nuclear localization signals were predicted using the cNLS mapper. The gene IDs of P. berghei PFG, *P. falciparum* PFG, and their homologs in Toxoplasma gondii, Eimeria tenella and Vitrella brassicaformis are PBANKA_0902300, PF3D7_1146800, TGGT1_239670, ETH2_1252400, and Vbra_10234, respectively.

(C) The amino acid sequences of Regions 1 and 2 from P. berghei PFG and its homologs from other apicomplexan parasites in (A) were aligned using the ClustalW program in MEGA X. The positions at which all these sequences have identical amino acids are indicated by two asterisks, and positions with amino acid residues possessing the same properties are indicated by one asterisk.

1. Figure 2: The Phenotype of Fd2 knockout should be characterized more comprehensively.It remains unclear whether ∆Fd2 parasite generate the same number of females but these are defective upon fertilization or whether there is also a decrease in the number of female gametocytes. Is the defect just post-fertilization and zygotes lyse or are there fewer fertilization events? If so is activation of female GCs effected?The number of male and female gametocytes should be quantified using sex-specific markers not affected by Fd2 knockout rather than providing a single image of each. The ability of ∆Fd2 GCs should also be evaluated.This is also important for the interpretation of Fig 2G. Is the down-regulation of the genes due to fewer female GCs or are the down-regulated genes only a subset of female-specific genes.

Response: In PFG(-) parasites, the rate of conversion into zygotes of female gametocytes decreased, and zygotes had lost capacity for developing into ookinetes. This indicates that gametocyte development (i.e., the ability to egress the erythrocyte and to fertilize) and zygote development were both impaired. This phenotype is consistent with the observation that genes expressed in female gametocytes are broadly downregulated. PFG is a TF, and its disruption led to decreased expression of hundreds of female genes. Thus, the observed phenotype may be derived from combined decreased expression of these genes. We believe further detailed phenotypic analyses will not generate much novel information on this TF. Instead, RNA-seq data in PFG(-) parasites and the targetome have promise in helping to characterize the functions of this TF.

1. Figure 3: what fraction of down-regulated genes have the Fd2 10mer motif?

Response: Thank you for the question. We investigated the upstream binding motifs of these genes. Of the 279 significantly down-regulated genes (containing 165 targets), 161 genes harbor the motif (including nine-base motifs that lack one lateral base which is likely not essential for binding) in their upstream regions (within 1,200 bp from the first methionine codon). However, this result has not been described in the revised manuscript because it is more important whether these regions harbor PFG peaks (upstream motifs can exist without being involved in the binding of PFG).

1. sAP2-FG (single) vs cAP2-FG (complex) nomenclature is confusing and possibly misleading since few TFs function in isolation and sAP2-FG likely functions in a complex that doesn't contain Fd2, possibly with another DNA binding protein that binds the TGCACA hexamer. The name for the distinct peaks should refer to the presence or absence of Fd2 in the complex, or maybe simply refer to them as complex A & B.

Response: As shown in the DIP-seq analysis results, AP2-FG can bind the motif by itself. In contrast, AP2-FG must form a complex with PFG to bind to the ten-base motif. The complex and single forms are named according to this difference (the presence or absence of PFG) and used solely in its relation with PFG. We wrote “In the following, we refer to the form with PFG as cAP2-FG or the complex form, and the form without PFG as sAP2-FG or the single form.” We believe that the nomenclature has sufficient clarity. However, we have partially (underlined) revised certain sentences in the discussion section as follows.

“As the expression of PFG increases via this mechanism, AP2-FG recruited by PFG (cAP2FG) increases and eventually becomes predominant in the transcriptional regulation of female gametocytes.”

“This suggests that the promoter of the CCP2 gene, which is a target of PFG only, is still active in AP2-FG(-)820 parasites.”

We recently reported that the TGCACA motif is a cis-activation motif in early gametocytes and important for both male and female gametocyte development. Thus we speculate that sAP2-FG is not involved in cis-activation by the TGCACA motif. The p-value of the six-base motif is indeed comparable to that of the five-base motif. However, the pvalue (calculated by Fisher’s exact test) in six-base motifs tend to be lower than that calculated in five-base motifs, because the population is much large. We speculate that there is a sequence-specific TF that may be expressed in early gametocytes and bind this motif, independently of AP2-FG.

1. I compared the overlap of peaks in the 4 ChIP-seq data sets:90% of the Fd2 peaks are shared with AP2-FG (binding 24% of shared peaks is lost in ∆AP2FG)10% are bound by Fd2 alone (binding at 35% of Fd2 is lost in ∆AP2-FG)75% of Fd2 peaks are bound independently of AP2-FG47% of AP2-FG peaks shared with Fd2 (binding at 71% of shared peaks is lost in ∆Fd2) 53% of AP2-FG peaks are bound only by AP2-FG (but binding at 82% of AP2-FG only peaks is still lost in the ∆Fd2)Binding at 78% of all AP2-FG peaks is lost in ∆Fd2This indicates that much of AP2-FG binding in regions even in regions devoid of Fd2 still depends on Fd2. What are possible explanations for this?
https://elife-rp.msubmit.net/eliferp_files/2023/04/03/00117573/00/117573_0_attach_10_17936_convrt.pdf

Response: In the ChIP-seq of AP2-FG in the absence of PFG, 441 peaks are still called. This means that at least 441 binding sites for AP2-FG independent of PFG exist. This is a straightforward conclusion from our ChIP-seq data. On the other hand, simple deduction of peaks between two ChIP-seq experiments (AP2-FG peaks minus PFG peaks) is not a precise method for determining sAP2-FG. Peak-calling is independently performed in each ChIP-seq experiment. Thus, peaks remaining after the deduction between two experiments can still contain peaks that are actually common, but which are differentially picked up through the process of peak calling. Even when using data obtained by the same ChIP-seq experiment, markedly different numbers of peaks are called according to the conditions for peak calling (in contrast, common peaks between two independent experiments increase the reliability of the data). If wanting to identify sAP2-FG peaks via comparisons between AP2-FG peaks and PFG peaks, the reviewer has to increase the number of PFG peaks by reducing the peak-calling threshold until the number of overlapping peaks between AP2-FG and PFG are saturated, and then deduce the overlapping peaks from the AP2-FG peaks. However, as described above, for the purposes of estimating the number of sAP2-FG, it would be better to perform ChIP-seq of AP2-FG in the absence of PFG.

1. Possible explanations of why recombinant Fd2 doesn't bind the TGCACA hexamer. It would also be good to note that the GCTCA AP2-FG motif found in Fig4G is now perfect match for the motif identified by protein binding microarray in Campbell et al.

Response: It is not known what sequence recombinant PFG binds. The TGCACA motif is not enriched in PFG peaks. If the reviewer is referring to AP2-FG, our findings that the recombinant AP2 domain binds the five-base motif strongly suggests that other TFs recognize this motif. As described in our response to comment 9, we recently reported that TGCACA is a cis-activating sequence important for the normal development of both male and female gametocytes. Therefore, we currently speculate that this motif is a binding motif of other TFs and is independent of AP2-FG.

We have mentioned the protein binding microarray data in the Results section as follows.

“The most enriched motif matched well with the binding sequence of the AP2 domain of *P. falciparum* AP2-FG, which was reported by Campbell et al.”

1. What might explain the strong enrichment for TGCACA in ChIPseq but when pulled down by AP2-FG DBD: another binding partner? requires more of AP2-DF than just DBD?

Response: As described above in our response to comment 6, we have recently submitted a preprint studying the roles of the remodeler subunit PbARID in gametocyte development. We reported that the remodeler subunit is recruited to the six-base motif and that the motif is a novel cis-activation element for early gametocyte development. We speculate that a proportion of AP2-FG targets are also targets of a TF that recognizes this motif and recruits the remodeler subunit. These two TFs may be involved in the regulation of early gametocyte genes but function independently.

1. Calling DNA pulldown with recombinant AP2-FG DNA-binding domain DNAImmunoprecipitation sequencing (DIP-seq) is confusing since there are no antibodies involved. Describing it directly as a pulldown of fragmented DNA will be clearer to the reader.

Response: Thank you for the comment. We have also recognized this discrepancy. However we called the method DIP-seq because the original paper reporting this method used this name, wherein it did not use antibodies to capture the MBP-fusion recombinant protein. Our experiment was performed using essentially the same methods, and thus we retained the name.

1. The legends and methods are very sparse and should include substantially more detail.

Response: Thank you for the comment. We have revised the description of the FACS experimental method for clarity.

1. BigWig files for all ChIPseq enrichment used for analysis in this study need to be provided.(two replicates each of : Fd2 in WT, Fd2 in ∆AP2-GF, AP2-FG in WT, AP2-FG in ∆Fd2)

Response: We have deposited the BigWig files to GEO (GSE.226028 and GSE114096).

1. Tables of ChIP data need to have both summits and peaks and need to list nearest gene. Also the ChIPseq peaks for Fd2 are surprisingly broad (ChIP peaks are very large, e.g. 68% of Fd2 peaks (dataset2) are greater than 1000kb) give its specificity for a long motif. Why is this?

Response: We have revised Table S2 to include the nearest genes. We are unsure why peaks in the over 1000-bp peak region exist in such high proportions. However, this proportion was also high in our previous ChIP-seq data. Therefore, we speculate that this is a tendency of peak-calling by MACS2. We did not use these values in this paper. For example, targets were predicted using peak summits, and binding motifs were calculated using the 100-base regions around peak summits.

1. Figure 5E: The positions of the 10mer and 5mer motifs in the promoter should be indicated as well as the length of the promoter. Moreover, mutation of just the 5bp motifs would be valuable to understand if 10mer is sufficient for expression of the reporter.

Response: Thank you for the comment. We have revised the figure accordingly. The majority of female-specific promoters only harbor ten-base motifs. Thus the ten-base motif is sufficient for evaluating reporter activity (i.e., it would function without five-base motifs).

1. How is AP2-FG expression affected in ∆Fd2 and vice versa?

Response: According to our previous microarray data, PFG expression was not significantly downregulated by disruption of AP2-FG. This may be because PFG transcriptionally activates itself through a positive feedback loop after being induced by AP2-G. Similarly, according to our present study, AP2-FG expression was not downregulated by PFG disruption. This may be because AP2-FG is transcriptionally activated by AP2-G.

1. The single cell data in Russell et al. could easily be used to indicate the order of expression.

Response: Determining the expression order of gametocyte TFs via the single cell RNA-seq data from Russel et al. is difficult, because only a small number of parasite cells were considered to be in the early gametocyte stage in this study. This is because the parasites were cultured for 24h before the analysis. The analysis suggested by the reviewer may be possible via single cell RNA-seq, but the experiments must be performed with more focus on the early gametocyte stage.

1. A discussion of the implication of *P. falciparum* transmission would be appreciated.

Response: Thank you for the comment. We have added the following to the Discussion section:

“*P. falciparum* gametocytes require 9-12 days to mature, which is much longer than that of P. berghei. Meanwhile, it has been reported that the ten-base motif is highly enriched in the upstream regions of female-specific genes also in *P. falciparum*. Thus, despite the difference in maturation periods, PFG is likely to play an important role in the transcriptional regulation of female *P. falciparum* gametocyte development."

1. The lack of identifiable DNA binding domains in Fd2 is intriguing given the strong sequence-specificity. Do the authors think they have identified a new DNA-binding fold ?Alphafold of the orthologs with contiguous regions 1&2 might offer insight.

Response: We speculate that these regions function as DNA binding domains. We performed analysis using Alfafold2 according to the comment. However, the predicted structure of the region was not similar to any other canonical DNA-binding domains. Thus, it may be a novel DNA-binding fold as the reviewer mentioned. Further studies such as binding assays using recombinant proteins would be necessary to confirm this, but thus far we have not successfully obtained the soluble proteins of these regions.